# *Populus euphratica* JRL Mediates ABA Response, Ionic and ROS Homeostasis in Arabidopsis under Salt Stress

**DOI:** 10.3390/ijms20040815

**Published:** 2019-02-14

**Authors:** Huilong Zhang, Chen Deng, Jun Yao, Yan-Li Zhang, Yi-Nan Zhang, Shurong Deng, Nan Zhao, Gang Sa, Xiaoyang Zhou, Cunfu Lu, Shanzhi Lin, Rui Zhao, Shaoliang Chen

**Affiliations:** 1Beijing Advanced Innovation Center for Tree Breeding by Molecular Design, College of Biological Sciences and Technology, Beijing Forestry University, Beijing 100083, China; hlzhang2018@126.com (H.Z.); ced501@163.com (C.D.); yaojun@bjfu.edu.cn (J.Y.); z585788@163.com (Y.-L.Z.); xhzyn007@163.com (Y.-N.Z.); danceon@126.com (S.D.); zhaonan19880921@126.com (N.Z.); sg_1214@126.com (G.S.); zhouxiaoyang@bjfu.edu.cn (X.Z.); lucunfu@bjfu.edu.cn (C.L.); szlin@bjfu.edu.cn (S.L.); 2State Key Laboratory of Tree Genetics and Breeding, The Research Institute of Forestry, Chinese Academy of Forestry, Beijing 100091, China

**Keywords:** jacalin-related lectin, *Populus euphratica*, NaCl, antioxidant enzyme, abscisic acid, K^+^/Na^+^ homeostasis, non-invasive micro-test technique

## Abstract

Sodium chloride (NaCl) induced expression of a jacalin-related mannose-binding lectin (*JRL*) gene in leaves, roots, and callus cultures of *Populus euphratica* (salt-resistant poplar). To explore the mechanism of the *PeJRL* in salinity tolerance, the full length of *PeJRL* was cloned from *P. euphratica* and was transformed into Arabidopsis. PeJRL was localized to the cytoplasm in mesophyll cells. Overexpression of *PeJRL* in Arabidopsis significantly improved the salt tolerance of transgenic plants, in terms of seed germination, root growth, and electrolyte leakage during seedling establishment. Under NaCl stress, transgenic plants retained K^+^ and limited the accumulation of Na^+^. *PeJRL*-transgenic lines increased Na^+^ extrusion, which was associated with the upward regulation of *SOS1*, *AHA1*, and *AHA2* genes encoding plasma membrane Na^+^/proton (H^+^) antiporter and H^+^-pumps. The activated H^+^-ATPases in *PeJRL*-overexpressed plants restricted the channel-mediated loss of K^+^ that was activated by NaCl-induced depolarization. Under salt stress, *PeJRL*–transgenic Arabidopsis maintained reactive oxygen species (ROS) homeostasis by activating the antioxidant enzymes and reducing the production of O_2_^−^ through downregulation of NADPH oxidases. Of note, the *PeJRL-*transgenic Arabidopsis repressed abscisic acid (ABA) biosynthesis, thus reducing the ABA-elicited ROS production and the oxidative damage during the period of salt stress. A schematic model was proposed to show the mediation of PeJRL on ABA response, and ionic and ROS homeostasis under NaCl stress.

## 1. Introduction

Plants are frequently challenged by various environmental stressors, which inhibit plant growth and crop production. Among these unfavourable environmental factors, salinity presents a serious threat to plant growth and development [1,2,3,4]. Salt stress leads to water deficiency and ion toxicity, which cause oxidative damage in plants [5,6]. In addition, high salt alters the expression level of stress-related genes that are involved in ionic homeostasis and anti-oxidant defense [7]. Accumulating evidence reveals that higher plants have developed signalling networks for sensing and adapting to salinity stress [8,9]. Lectins are able to bind carbohydrates [10,11] and play an important role in plant adaptation to adverse environments [12]. It has been shown that plant lectins inhibit infection of pathogen fungi and insects [13,14,15]. In wheat, jacalin-related lectin (JRL) genes such as *TaJRL2.1*, *TaJRL53*, *TaJRL18.2*, *TaJRL2*7, *TaJRL43*, and *TaJRL48* are induced in inflorescence after *Fusarium graminearum* inoculation, and *TaJRL55* is induced by Hessian fly infection [16]. *Allium sativum* ASAL, a novel garlic lectin, is toxic towards hemipteran pests in *ASAL*-transgenic plants [14]. The snowdrop lectin GNA (*Galanthus nivalis* agglutinin) confers resistance to sap-sucking insects, e.g., brown planthopper (*Nilaparvata lugens*) [13]. Xiang et al. (2011) suggest that TaJRLL1 serves as a signalling component in the jasmonate (JA)- and salicylic acid (SA)-dependent defence pathways [15]. *TaJRLL1*-transformed *Arabidopsis thaliana* were shown to be resistance to *Botrytis cinerea* and *Fusarium graminearum* [15]. In addition to biotic stress, several lectin genes also exhibit different expressions under abiotic stress [16,17,18]. Digital expression analysis (DEA) has shown that 25 *TaJRL* genes were responsive to abiotic stress, such as water shortage, low temperature, aluminium, and salt stress [16]. A rice (*Oryza sativa* L.) lectin was isolated from salt-stressed plants and previously characterized by Zhang et al. (2000) [19]. Recently, *OsJRL* overexpression resulted in enhanced tolerance to NaCl in *Escherichia coli* and rice [18]. The OsJRL is also suggested to be involved in stress signal transduction and cell protection in rice [18]. However, the regulatory roles of *Populus* JRL family genes in salt tolerance are not yet fully understood in trees.

The phytohormone abscisic acid (ABA) is crucial for plant adapting to unfavourable environmental conditions [20,21,22,23]. High salinity and drought dramatically increase the ABA level, which in turn induces the expression of many genes involved in stress responses [24]. Abscisic acid has been shown to increase the synthesis of lectin in wheat cell cultures [25]. Furthermore, *OsJRL* transcription is induced in response to ABA treatment in *O. sativa* subspecies japonica “Nipponbare” [18]. It is interesting to find that overexpression of *OsJRL* resulted in ABA sensitivity in rice [18]. It is suggested that *OsJRL* overexpressing in rice strengthened the salt stress signal; this would increase ABA sensitivity, thus leading to increased transcription of downstream salt-responsible genes [18]. At present, the sensitivity to ABA and the relevance to salt tolerance have scarcely been investigated in plants overexpressing *JRL* from tree species.

*Populus euphratica* has been considered as a salt-resistant model species to address stress physiology in woody plants [8]. *Populus euphratica* retains a greater capacity to control reactive oxygen species (ROS) and K^+^/Na^+^ homeostasis under saline conditions [5,6,7,8,26,27,28,29,30,31,32]. The signalling network of *P. euphratica* upon salt stress has been extensively investigated in previous studies [33,34,35]. In the present study, we observed that salt treatment increased *PeJRL* expression in *P. euphratica* leaves, roots, and callus cells. To determine whether PeJRL contributed to salt tolerance, *PeJRL* was cloned from *P. euphratica* and introduced into Arabidopsis, the model plant. We examined ion relations, ROS production, activity of antioxidant enzymes, and transcription of encoding genes, such as superoxide dismutase (SOD), catalase (CAT), and peroxidase (POD) under NaCl stress. The aim was to evaluate the contribution of PeJRL in ionic homeostasis control and anti-oxidative defence in transgenic plants. In addition, the sensitivity to ABA in Arabidopsis overexpressing *PeJRL* and the relevance to salt tolerance were investigated in the present study. Our data showed that *PeJRL* overexpression increased plants’ ability to retain K^+^ and Na^+^ homeostasis during the observation period of NaCl treatment. Moreover, the *PeJRL*-overexpressing-plants maintained ROS homeostasis by activating the antioxidant enzymes and repressing ABA biosynthesis, thus reducing the ABA-elicited ROS production and the oxidative damage during the period of salt stress.

## 2. Results

### 2.1. Expression Profile of PeJRL upon Salt Exposure in Populus euphratica

To determine whether *PeJRL* was responsive to salt stress in *P. euphratica*, real-time quantitative PCR (RT-qPCR) was used to examine *PeJRL* transcript levels in callus cells, leaves, and roots. Exposure to NaCl (125–200 mM) significantly increased *PeJRL* transcript levels relative to the reference gene in *P. euphratica*, although the pattern varied between callus, leaves, and roots (Figure 1). In callus cells, the transcripts of *PeJRL* rapidly increased up to 3.59 fold after 12 h of salt treatment, then reached 7.72 fold at 24 h (Figure 1A). In *P. euphratica* leaves, *PeJRL* transcript levels gradually increased upon salt exposure and reached the maximum after 6–12 h of salt treatment (Figure 1B). The *PeJRL* transcript abundance was maintained at relatively high levels (2–5 fold) in the following time of salinity (24–72 h; Figure 1B). In *P. euphratica* roots, *PeJRL* transcript levels increased rapidly at 3 h after salt treatment and reached peak levels until 6 h, and then returned to the pre-treatment level in the following hours (Figure 1C). The drop of expression showing at leaves (24 h) and roots (12 h) implied the recovery of whole-plant water status after salt shock [29,36]. The following increase of *PeJRL* transcript in leaves at 48–72 h was presumably the result of plant response to build-up of salt ions that translocated from roots to shoots. Collectively, these results demonstrated that *PeJRL* was NaCl-inductive in this salt-resistant poplar.

### 2.2. PeJRL Protein Sequence and Phylogenetic Analysis

Sequence and phylogenetic analyses were carried out to determine whether PeJRL belongs to jacalin-related mannose-binding lectins. The 1338-bp full-length cDNA of *PeJRL* was cloned from *P. euphratica* leaves. It encoded a putative protein of 445 amino acids (Figure 2A). The protein has a predicted molecular weight of 47.18 kDa with an iso-electric point (pI) 5.94 (Figure 2A). The PeJRL amino acid sequence displays higher similarity to *P. trichocarpa* JRL (PtJRL) than that of other higher-order plants. The PeJRL protein contains a conserved C terminal sequence, and three typical jacalin domains which are sugar-binding protein domains mostly found in higher plants (Figure 2A). Figure 2B shows the phylogenetic relationship between *PeJRL* and *JRLs* from other plant species. The constructed phylogenetic dendrogram shows the evolutionary conservation of *PeJRL* to other jacalin-related mannose-binding lectins (Figure 2B).

### 2.3. Subcellular Localization of PeJRL

Jacalin-related mannose-binding lectin has been shown to be a nucleocytoplasmic lectin in plants [18]. To visualize PeJRL localization within the cell, green fluorescent protein (GFP) was fused to the full-length *PeJRL* cDNA and transiently expressed in Arabidopsis protoplasts. Confocal laser scanning microscopy showed that PeJRL localized to the cytoplasm in the mesophyll cells (Figure 3A). However, the protoplasts transformed with GFP alone showed GFP fluorescence throughout the cytoplasm and nucleus (Figure 3B).

### 2.4. Overexpression of PeJRL in Arabidopsis

To elucidate the function of *PeJRL* in salt tolerance, the ORF of *PeJRL* was transformed into wildtype (WT) Arabidopsis under the control of *CaMV 35S* promoter. Six independent transgenic lines, i.e., OE6, 7, 12, 18, 21, and 23 were obtained. Semi-quantitative reverse transcription PCR (RT-PCR) and RT-qPCR results showed that OE7 and OE23 had higher transcript levels of *PeJRL* than the other lines (Figure 4A,B). Therefore, these two lines were selected for subsequent phenotype tests.

### 2.5. PeJRL Overexpression in Arabidopsis Enhanced Salt Tolerance

To determine whether PeJRL affected salt tolerance during seedling establishment, seeds of wild-type (WT), vector control (VC), and transgenic plants were germinated on 1/2 Murashige–Skoog (MS) medium supplemented with increasing NaCl (75–125 mM for 10 d, Figure 5A). The two transgenic lines performed much better than WT Arabidopsis and VC at 125 mM NaCl, in terms of leaf opening and greening (Figure 5A). Root length was inhibited upon NaCl exposure, but the growth was less suppressed in transgenic lines at the tested concentrations (75 or 125 mM), as compared to WT and VC plants (Figure 5B,C). 

Electrolyte leakage (EL) is an important indicator reflecting the membrane permeability and lipid peroxidation under stress conditions [5]. Electrolyte leakage was examined to determine whether the salt treatment disrupted the plasma membrane. Electrolyte leakage detected in all lines increased with the increasing NaCl concentrations (Figure 5D). There were significant differences in EL among WT, VC, and transgenic lines after exposure to 75 or 125 mM NaCl. The *PeJRL*-overexpressed lines exhibited significantly lower EL than WT and VC under salt stress (Figure 5D), indicating that the membrane integrity of transgenic plants was less disrupted by lipid peroxidation.

### 2.6. Na^+^ Concentrations in Roots

It has been previously shown that salt-tolerant plants could control Na^+^ uptake and avoid an excessive accumulation [20,26,28]. To determine whether PeJRL contributed to controlling the Na^+^ accumulation, Na^+^ concentrations within root cells were detected with a Na^+^ specific probe, CoroNa™ Green [35]. The concentrations of Na^+^ significantly increased after 12 h of salt treatment (Figure 6). The fluorescence intensity in WT Arabidopsis and VC was greater than that observed in OE7 and OE23 (Figure 6B). The specific fluorescence intensity was almost non-detectable under control conditions (Figure 6A), indicating that the Na^+^ concentration was very low in the roots of all tested genotypes. 

### 2.7. Na^+^, K^+^, and H^+^ Fluxes in Root Tips and Transcription of K^+^/Na^+^ Homeostasis Genes

The ability to retain K^+^/Na^+^ homeostasis is critical for plant adaptation to salt stress [31,32,33,34]. To determine whether PeJRL contributed to maintaining the ionic homeostasis, the root ionic fluxes were examined by non-invasive micro-test technique (NMT) in all test lines. Non-invasive micro-test technique recording of roots showed that NaCl caused an increased efflux of Na^+^ in all tested lines (Figure 7A). It is worth noting that transgenic plants exhibited three-fold higher Na^+^ efflux than WT and VC after 12 h of NaCl stress (Figure 7A). In all tested roots, the influx of H^+^ was shifted to efflux by NaCl treatment (Figure 7B). However, the H^+^ effluxes in transgenic lines were notably higher than in WT Arabidopsis and VC (2-fold, Figure 7B). NaCl exposure increased the efflux of K^+^ in all tested lines, while transgenic plants exhibited lower K^+^ loss than in WT Arabidopsis and VC (Figure 7C). 

The Na^+^ extrusion and K^+^ maintenance depends on the activity of Na^+^/H^+^ antiporters and H^+^-pumps in the plasma membrane (PM) [8,33]. The transcript abundance of K^+^/Na^+^ homeostasis-related genes encoding plasma membrane H^+^-ATPases (*AtAHA1* and *AtAHA2*), Na^+^/H^+^ antiporter (*AtSOS1*), and high-affinity K^+^ transporter (*AtHKT1*) were compared under NaCl stress. The expression of *AtAHA1*, *AtAHA2*, *AtSOS1*, and *AtHKT1* were significantly higher in transgenic plants than in the WT and VC (Figure 7D).

### 2.8. Anti-Oxidative Enzyme Activity and Transcript Levels of Encoding Genes

Salt stress-elicited ROS causes oxidative damage to plant cells [5,6]. It is critically important to enhance ROS-scavenging capacity by increasing the activity of anti-oxidative enzymes under saline conditions [5,6]. To determine whether *PeJRL* overexpression could affect activities of anti-oxidative enzymes, SOD, POD, and CAT activities were measured in all tested lines under salt treatment. Transgenic lines retained significantly higher activity of these antioxidant enzymes than WT Arabidopsis and VC (Figure 8A,C,E). The gene expression of *SOD*, *POD*, and *CAT* displayed a high similarity to the pattern of their activity profiles under NaCl stress. Transgenic plants retained higher transcription levels of *AtSOD*, *AtCAT*, and *AtPOD* relative to the WT Arabidopsis and VC (Figure 8B,D,F).

### 2.9. H_2_O_2_ Levels in Root Cells and Transcription of AtRBOHD and AtRBOHF

The concentration of H_2_O_2_ could reflect the capacity of ROS-scavenging capacity in salt-stressed plants [5,6]. Hydrogen peroxide production was examined to determine whether *PeJRL*-transgenic plants could control ROS homeostasis under NaCl stress. The salt treatment significantly increased H_2_O_2_ levels in WT, VC, and *PeJRL*-transgenic roots (Figure 9A). However, salinized transgenic plants retained obviously lower H_2_O_2_ levels than that in WT Arabidopsis and VC (Figure 9A). In accordance, transgenic lines remained lower transcription of *AtRBOHD* and *AtRBOHF* genes, encoding Arabidopsis reduced nicotinamide adenine dinucleotide phosphate (NADPH) oxidases, under salt treatment (Figure 9B,C). The transcripts of *AtRBOHD* were similar in no-salt control plants of all tested lines, although *PeJRL*-transgenic lines exhibited lower *AtRBOHF* transcript than WT and VC (Figure 9B,C).

### 2.10. PeJRL Overexpression Increased ABA Sensitivity in the Presence and Absence of Salt

It is suggested that a rice JRL enhances salt stress signalling by increasing the sensitivity to ABA [18]. To determine whether PeJRL is involved in the ABA signalling pathway, the sensitivity to ABA was examined in *PeJRL*-transgenic plants. In the absence of NaCl, overexpression of *PeJRL* increased the sensitivity to ABA in Arabidopsis. Exogenous ABA, at a dose of 0.6 or 5 µM, caused a more pronounced reduction of plant growth and root length in *PeJRL*-transgenic lines, compared with the WT Arabidopsis and VC (Figure 10A–C). Abscisic acid at the tested doses (0.6 or 5 µM) increased EL in all tested lines, but a more pronounced enhancement was observed in transgenic lines (Figure 10D). 

In the presence of NaCl, ABA increased H_2_O_2_ content in all tested lines (Figure 11A). In comparison, the ABA-elicited H_2_O_2_ increase was more pronounced in transgenic lines, because these plants retained a lower level of H_2_O_2_ relative to WT and VC in non-ABA treatment (Figure 11A). Abscisic acid significantly increased transcription of *AtRBOHD* and *AtRBOHF* genes in transgenic lines, which is in contrast to non-ABA conditions where it remained typically lower in transgenes than in WT and VC (Figure 11B).

### 2.11. PeJRL Downregulated Synthesis of Endogenous ABA under NaCl

Under NaCl stress, abscisic acid (ABA) content in *PeJRL*-overexpressed plants was significantly lower than in WT Arabidopsis and VC (Figure 12A). In accordance, *PeJRL*-transgenic plants downregulated the expression of *AtNCED2* and *AtNCED9*, two key enzymes in the biosynthesis of ABA under salt treatment (Figure 12B).

### 2.12. PeJRL Altered Expression Profile of ABA-Mediated Salt Stress Responsive Genes

To investigate whether PeJRL interacts with the ABA signalling network under salt stress, the expression levels of a subset of ABA-mediated salt stress responsive genes were examined in transgenic plants. These genes include *ABF4/AREB2* [37], *ABI5* [38], *DREB1A, DREB2A* [39], *MYB2* [40], *RD29A* [41], *RAB18* [42], *SnRK2.2*, and *SnRK2.3* [24,43]. Most of the tested genes such as *DREB1A, DREB2A, MYB2, RD29A*, and *SNRK2.2* showed an increased transcript in *PeJRL*-transgenic lines (OE7 and OE23), compared with those in WT and vector controls (Figure 13). A few of the ABA-mediated stress-responsive genes (*ABF4, ABI5, RAB18*, and *SnRK2.3*) remained unchanged or downregulated in transgenic plants (Figure 13). The upregulated *DREB, MYB, RD29* and *SNRK* may increase the plant sensitivity to ABA. 

## 3. Discussion

The present study confirmed the novel role of jacalin-related mannose-binding lectin in the salt response of *P. euphratica*. The PeJRL protein was shown to localize to the cytoplasmic region (Figure 3). Unexpectedly, we could not detect the nuclear localization, although PeJRL sequence showed a high similarity to orthologs of nucleocytoplasmic lectins in other plant species (Figure 2) [11]. Transcription analyses indicated that *PeJRL* was upregulated by NaCl in *P. euphratica* calli, leaves, and roots (Figure 1). Similarly, *TaJRL26* was responsive to salt stress in wheat roots [16]. Salinity-stress induces the synthesis of a mannose-specific lectin in rice plants [19]. *PeJRL* overexpression in Arabidopsis significantly increased salt tolerance of transgenic plants, in terms of plant performance, primary root length, and membrane leakage during seedling establishments (Figure 4, Figure 5). Thus, it can be inferred that *PeJRL* contributes to salt adaptation in *P. euphratica*. Similarly, *OsJRL* overexpression in rice enhanced salt tolerance of transgenic plants [18]. Our data showed that the enhanced tolerance of Arabidopsis plants overexpressing *PeJRL* mainly resulted from the increased ability to regulate K^+^/Na^+^ and ROS homeostasis. 

In *PeJRL*-overexpressed Arabidopsis, the enhancement of NaCl tolerance relied on, at least in part, the ability for Na^+^ extrusion and K^+^ maintenance (Figure 6 and Figure 7). There was less Na^+^ accumulation in root cells of transgenic plants (Figure 6) due to a higher ability for extruding Na^+^ in root tips (Figure 7A). Na^+^ extrusion is critical for plants adapting to salt stress environment [7,8,20,28,30,32,33,34,44]. Furthermore, we observed an apparent H^+^ efflux in these salt-exposed roots (Figure 7C). This indicates the activated H^+^-ATPases could establish an H^+^ gradient across the PM, thus promoting the exchange of Na^+^ with H^+^ [45]. In accordance, the expression of *AtSOS1*, *AtAHA1*, and *ATAHA2* genes encoding PM Na^+^/H^+^ antiporters and H^+^-ATPases were upward regulated in transgenic plants (Figure 7D). Therefore, in *PeJRL*-transgenic lines, the ion flux profiles and transcription data suggest that the H^+^-pumps contributed to Na^+^ extrusion via increasing the antiport of Na^+^/H^+^ across the plasma membrane [30,31,46]. In addition, the upregulation of *AtHKT* under salt stress might also avoid excessive build-up of Na^+^ in transgenic plants (Figure 7D). Under salt stress Arabidopsis high-affinity K^+^ transporters (HKT) have been shown to mediate retrieval of Na^+^ from the xylem, thus preventing excessive Na^+^ in leaves [47,48,49,50,51,52]. Similarly, *OsJRL*-transgenic rice maintained higher transcript levels of three rice *HKT* genes (*HKT1;3*, *HKT1;4* and *HKT1;5*) than the wild type under salinity treatment [18]. Collectively, these results showed that overexpression of *JRL* genes could protect plant cells against Na^+^ excess through upregulation of Na^+^ transporter genes.

Furthermore, *PeJRL*-overexpression enabled transgenic plants to reduce the loss of K^+^ under saline conditions, as compared to WT (Figure 7). It has been repeatedly shown that salt tolerance in plants is related to the capacity for maintaining K^+^ homeostasis [30,33,34]. Salt-elicited loss of K^+^ is through the K^+^ channels activated by PM depolarization [30,53,54]. In *PeJRL*-overexpressed plants, the activated H^+^-ATPases restricted the membrane depolarization, thus reducing the channel-mediated K^+^ loss under salt stress [30].

Under NaCl stress, *PeJRL*-transgenic plants displayed significantly lower H_2_O_2_ content than WT and VC (Figure 9). This is mainly the result of increased activity of CAT and POD, the main antioxidant enzyme scavenging H_2_O_2_ (Figure 8). In accordance, transcription of *CAT* and *POD* was increased upon NaCl exposure in transgenic plants (Figure 8). The expression and activity of SOD significantly increased upon salt exposure, similar to the trend of CAT and POD (Figure 8). Superoxide dismutases are considered to be the first defence line against O_2_^−^ and the reaction product [55,56]. In the present study, the coincident increase of SOD with CAT and POD in *PeJRL*-transgenes reveals an elevated capacity to detoxify both O_2_^−^ and H_2_O_2_ that were caused by NaCl, which is required for rapid removal of ROS and thus avoids oxidative damage [8]. In addition, we observed that *PeJRL*-overexpression limited the salt-induced expression of *AtRBOHD* and *AtRBOHF* (Figure 9). This indicates that the transgenic plants could control the production of O_2_^−^ via downregulation of NADPH oxidases under salt stress.

Unexpectedly, *PeJRL* overexpression lowered ABA content in salt-stressed Arabidopsis (Figure 12). This was due to the low transcripts of *AtNCED2* and *AtNCED9* (Figure 12), the two key enzymes in ABA biosynthesis [57,58,59]. It is noting to find that PeJRL increased ABA sensitivity in Arabidopsis in terms of plant performance, root length, and EL (Figure 10). Similarly, He et al. (2017) confirmed the increased ABA sensitivity in *OsJRL* transgenic rice [18]. We observed that a variety of ABA-mediated stress-responsive genes, such as *DREB1A, DREB2A, MYB2, RD29A* and *SNRK2.2*, showed an increased transcript in *PeJRL*-transgenic lines (Figure 13). The is in agreement with He et al. (2017), who found that ABA increased expression of genes encoding rice late embryogenesis abundant proteins (*LEA19a*, *LEA23*, and *LEA24*), HKT (Na^+^ transporters, *HKT1;3*, *HKT1;4*, and *HKT1;5*), and DREB (*DREB1A* and *DREB2B*) [18]. Thus, it can be inferred that JRL interacts with ABA signalling network to regulate plant response to salt stress. Abscisic acid suppressed shoot length of rice plants, and the inhibition was more pronounced in *OsJRL*-overexpressing seedlings than wild type [18]. In our study, the ABA-induced increase of EL in transgenic Arabidopsis was presumably related to H_2_O_2_ production. Under NaCl salinity, ABA-induced increase of H_2_O_2_ was more pronounced in *PeJRL*-overexpressed plants (2.9- and 2.3-fold), as compared to WT and VC (1.2- and 1.4-fold) (Figure 11). This is consistent to the drastic rise of *AtRBOHD* and *AtRBOHF* in salinized transgenic plants, while the expression of NADPH oxidase was less affected by combined treatment with ABA and NaCl in wild-type plants (Figure 11). Our data suggest that ABA-induced NADPH oxidase contributed to H_2_O_2_ production in salt-stressed plants. Therefore, *PeJRL*-overexpressed plants decrease the ABA biosynthesis under NaCl stress, this is beneficial for salinized plants to control ROS homeostasis since NaCl-elicited ABA caused a marked rise of H_2_O_2_.

## 4. Materials and Methods

### 4.1. Plant Materials and Treatments

*Populus euphratica* seedlings (one-year-old, from the Xinjiang Uygur Autonomous Region, China) were cultured in a greenhouse at the experimental nursery of Beijing Forestry University as described previously [30,31]. Seedlings were raised for two months, then salinized with NaCl (200 mM, 2 L) in Hoagland’s nutrient solution (full-strength). Leaves were sampled from upper shoots at 0, 1, 3, 6, 12, 24, 48, and 72 h. These leaves were immediately frozen in liquid N_2_, stored at a freezer (–80 °C) for real-time quantitative PCR (RT-qPCR) assays.

*Populus euphratica* callus cultures were induced and sub-cultured every 15 days in the dark at 25 °C [33,34]. Ten days after transplantation onto new MS medium, *P. euphratica* cultures were subjected to 125 mM NaCl in liquid MS medium, and controls were cultured in liquid MS medium supplemented without NaCl. Calluses were harvested after 0, 3, 12, and 24 h of salt treatment, frozen, and stored at a freezer (−80 °C) for subsequent RNA isolation and RT-qPCR.

Surface-sterilized seeds of *Arabidopsis thaliana* (Columbia, Col-0) were plated on MS solid medium (half-strength) with the addition of sucrose (1%, *w*/*v*) and Phytagel (0.24%, Sigma–Aldrich). The seeds were stratified at 4 °C for 3 days before transferring to growth chambers with a 16 h photoperiod (150 µmol m^−2^·s^−1^ irradiation, 21 °C, 80% relative humidity). Seven days after germination, seedlings were transplanted and grown at compost soil under a 16 h photoperiod in a growth chamber.

### 4.2. Full-Length PeJRL Gene Cloning and Sequence Analysis

Total RNA of *P. euphratica* leaves was isolated with EASYspin Plus Plant RNA Kit (AidLab, Beijing, China) based on the protocol instructions of manufacturer. The RNA (1 μg) was used as template to synthesize first-strand cDNA with oligo (dT) primer (Promega, Madison, WI, USA) and M-MLV Reverse Transcriptase (Promega, Madison, WI, USA) [45]. Full-length cDNA sequence of *PeJRL* was amplified with specific forward and reverse primers: 5′-ATGGCATCCTTGGAACGAATC-3′, 5′-TTAGATTGTCGTCTCTGGTTTGAC-3′. The gel-purified PCR products were then ligated to the pMD18-T (Takara, Kusatsu, Japan) vector for DNA sequencing.

We compared the amino acid sequences of JRLs from different plant species with ClustalW (http://www.genome.jp/tools/clustalw/, accessed on: 24 September 2018). Phylogenetic tree of JRL was constructed by the neighbour-joining method with 1000 bootstrap replicates using MEGA 5.2 software (http://www.megasoftware.net/index.php, accessed on: 24 September 2018). 

### 4.3. Subcellular Localization Analysis

Full-length cDNA was inserted into the modified pGreen0029-GFP vector between the HindIII and XmaI sites. *PeJRL* was fused upstream of the reporter gene, green fluorescent protein (GFP), under the control of the *CaMV35S* (cauliflower mosaic virus 35S) promoter. Protoplasts of Arabidopsis leaves were isolated and transformed using polyethylene glycol (PEG)-mediated protocol as per Yoo et al. (2007) [60]. The intensity of fluorescence was examined after incubation at 21 °C for 16–20 h. Confocal images were obtained with a confocal laser scanning microscope (Leica Microsystems GmbH, Wetzlar, Germany) at 510–535 nm (for GFP fluorescence) and 650–750 nm (for chlorophyll fluorescence), respectively. The confocal parameters were set as described in previous studies: excitation wavelength was 488 nm, and emission wavelength was 610–700 nm [61].

### 4.4. Overexpression of PeJRL in Arabidopsis

The full-length cDNA of *PeJRL* was cloned into pMDC85—the plant expression vector. Transcription of *PeJRL* was driven by the *CaMV35S* promoter. *PeJRL*: pMDC85 construct was transferred into *Agrobacterium tumefaciens* strain, GV3101, and then transformed into wild-type *A. thaliana* (Col-0 ecotype) [62]. The blank pMDC85 vector was introduced into wild-type *A. thaliana* (Col-0 ecotype) plants as a vector control (VC). Putative transgenic lines were screened on MS medium (half strength) supplemented with 25 mg/L hygromycin. To testify the transcription levels of the target gene, six homozygous lines (third generation) were selected according to the segregation ratio and verified by semi-quantitative reverse transcription PCR and RT-qPCR using *PeJRL*-specific primers (Appendix A). Two independent transgenic lines, OE7 and OE23, which exhibited higher transcript abundance of *PeJRL*, were used for salt experiments.

### 4.5. Quantitative PCR

The expression of *PeJRL* in *P. euphratica* and transgenic Arabidopsis were analysed using RT-qPCR and semi-quantitative reverse-transcription PCR (RT-PCR), respectively. Total RNA was isolated respectively from control and NaCl-treated *P. euphratica* leaves and Arabidopsis plants. The RNA isolation was performed as mentioned above. The RNase-free DNase (Promega, Madison, WI, USA) was used to eliminate the DNA in isolated RNA. The protocols for RT-PCR and RT-qPCR amplification were followed as per Han et al. [63] (2013) and Zhang et al. [64] (2017). House-keeping genes, *AtACTIN2* and *PeACTIN7*, were used as internal controls for Arabidopsis and *P. euphratica*. Specific primers for target and house-keeping genes were listed in Appendix A. Quantitative PCR amplifications were carried out in triplicate for each sample and experiments were repeated at least three times. The Ct value of the target gene was normalized with the 2^−ΔΔC^^T^ method [65].

### 4.6. Screening Tests for Salt Tolerance

To determine germination efficiency for WT, VC, and transgenic lines, approximately 70 seeds for each genotype were sterilized with NaClO (5%, *v*/*v*) for 10 min, then washed with sterile water 5 times. Seeds were plated on MS medium supplemented with 0, 75, or 125 mM NaCl. These seeds were incubated at 4 °C for 3 days, then transferred to a growth chamber (21 °C) under a 16 h/8 h (white light/dark) photoperiod. The seedling establishment was examined after 10 days of salt treatment. 

Five-day-old seedlings of WT, VC, and transgenic lines grown on half-strength MS medium were used for the root-growth assay. These seedlings were treated with 0, 75 or 125 mM NaCl on MS medium. After 10 days, primary root length was measured with ImageJ (Version 1.48, National Institutes of Health, Bethesda, Rockville Pike, MD, USA).

### 4.7. Measurements Leaf Electrolyte Leakage 

Relative electrolyte leakage was measured using fresh leaves sampled from control and salinized plants. The collected leaves were placed in an Eppendorf tube and incubated in 4 mL of deionized water, then electrical conductance (EC) measured after 24 h incubation at room temperature. The final electrical conductance was measured after the samples were boiled for 30 min to induce complete leakage and cooled to room temperature [61]. The leaf electrolyte leakage was calculated followed the formula as follows:EL (%) = (EC1/EC2) × 100%
where the EC1 is the initial electrical conductance of the leaves, EC2 represents the final electrical conductance.

### 4.8. Detection of Na^+^ Concentrations in Roots

The concentrations of Na^+^ in roots were detected with a sodium-specific dye, CoroNa-Green AM (Invitrogen, Carlsbad, CA, USA). Seedlings of WT, VC, and transgene lines (seven-day-old) were salinized with NaCl (0 or 125 mM) for 12 h in 1/2 MS liquid solution. Then roots were stained with the Na^+^ probe in the dark for 2 h, followed by washing with redistilled water (3–4 times). The fluorescence of CoroNa in root cells was visualized with a confocal microscope (Leica SP5, Microsystems GmbH, Wetzlar, German) with excitation wavelength at 488 nm, and emission wavelength at 510–530 nm. The fluorescent intensity was calculated using ImageJ (Version 1.48, National Institutes of Health, Bethesda, Rockville Pike, MD, USA) [33,34,35,45].

### 4.9. H_2_O_2_ Measurement and Expression of NADPH Oxidase

The H_2_O_2_-specific fluorescent probe, H_2_DCF-DA (2′,7′-dichlorodihydro-fluorescein diacetate; Molecular Probe, Eugene, OR, USA), was prepared in half-strength MS liquid solution (pH 5.8) and used to detect H_2_O_2_ level in roots. Seedlings of WT, VC and transgenic lines (seven-day-old) were salinized without or with 125 mM NaCl for 12 h. Then, roots were stained with H_2_DCF-DA (10 µM) in the dark for 5 min. Thereafter, these H_2_DCF-DA-loaded plants were washed three to four times with half-strength MS liquid solution. Fluorescence of H_2_DCF-DA was detected with a Leica SP5 confocal microscope with excitation at 488 nm and emission at 510–530 nm [33,34,35]. 

The expression level of NADPH oxidase gene, *AtRBOHD* and *AtRBOHF*, was examined by RT-qPCR using the primers listed in Appendix A.

### 4.10. Root Flux Recordings and Transcription of Ion Homeostasis-Related Genes

Wild-type, VC, and transgenic plants were grown on half-strength MS solid medium for seven days. Then these seedlings were transferred to MS medium and salinized without or with 125 mM NaCl for 12 h. Then roots were sampled and subjected to 30 min of equilibration in measuring solution containing the following components in mM: KCl (0.5), CaCl_2_ (0.1), MgCl_2_ (0.1), NaCl (0.1), and 2.5% sucrose (pH of solution was adjusted to 5.8). Afterwards, steady-state fluxes of Na^+^, K^+^, and H^+^ were measured continuously for 10 min at the meristematic zone (approximately 200 µm from the tip). The meristematic zone usually exhibited larger flux rates than mature zones [66,67]. In this study, root flux rates were recorded with the Non-invasive Micro-Test Technique (NMT-YG-100, Younger USA LLC, Amherst, MA, USA) with ASET 2.0 (Sciencewares, Falmouth, MA, USA) and iFluxes 1.0 (Young-erUSA, LLC, Amherst, MA, USA) software, which is capable of integrating and coordinating differential voltage signal collection, motion control and image capture simultaneously. The ionic flux was measured by shifting the ion-selective microelectrode between two sites close the roots over a pre-set length (30 μm for intact roots in this experiment) at a frequency in the range of 0.3–0.5 Hz. Ionic flux was calculated by Fick’s law of diffusion: *J* = −*D(dc/dx)*, where *J* represents the ionic flux in the *x* direction, *dc/dx* is the ionic concentration gradient, and *D* is the ionic diffusion constant [30,31,45,66,67,68]. The expression levels of K^+^/Na^+^ homeostasis-related genes encoding plasma membrane H^+^-ATPases (*AtAHA1* and *AtAHA2*), Na^+^/H^+^ antiporter (*AtSOS1*), and high-affinity K^+^ transporter (*AtHKT1*) were analysed by RT-qPCR using the primers listed in Appendix A.

### 4.11. Activity Measurements and Transcriptional Assays of Antioxidant Enzymes

For the enzymatic assays, transgenic seedlings, VC, and wild-type Arabidopsis (0.2 g) were frozen in liquid N_2_, ground to a fine powder, and then homogenized in 2 mL ice-cold potassium phosphate buffer (50 mM, pH 7.0) supplemented with poly(vinylpyrrolidone) (PVP) (1%), ethylenediaminetetraacetic acid (EDTA) (1 mM), and ascorbate acid (AsA) (1 mM). After centrifugation at 12,000× *g* for 20 min (at 4 °C) the supernatants were used to measure the activity of antioxidant enzymes, including superoxide dismutase (SOD), catalase (CAT), and peroxidase (POD). Total activity of these enzymes was determined as described previously [6,69,70,71,72]. In addition, the expression of *CAT*, *POD*, and *SOD* were analysed by RT-qPCR (Appendix A).

### 4.12. Determination of ABA Content and Expression of ABA Biosynthesis Genes

Five-day-old seedlings of transgenic, VC, and wild-type Arabidopsis were transferred to MS medium containing 75 mM NaCl for 10 days. Then Arabidopsis seedlings were sampled, frozen in liquid nitrogen, and homogenized in phosphate buffered saline (PBS, pH7.4). The homogenate was centrifuged at 3000× *g* for 20 min, and the supernatants were used to detect the level of endogenous ABA. Endogenous ABA level was measured in the supernatant using ELISA (MLBIO, Shanghai, China). The expression levels of ABA biosynthesis genes, *AtNCED2* and *AtNCED9*, were analysed by RT-qPCR (Appendix A).

### 4.13. ABA Tests for Transgenic Lines

Seed germination and seedling establishment for transgenic lines and wild-type were determined under ABA treatment. In brief, seeds for each genotype were sterilized with 5% NaClO (*v*/*v*) for 10 min, then washed with sterile water for 5 times. Seeds were germinated on MS medium containing 0, 0.6, or 5 µM ABA. After seed stratification, the seeds were transferred to a growth chamber (21 °C) under a 16 h/8 h (white light/dark) photoperiod, and then plant performance was examined after 10 days of ABA treatment. For root length and EL measurements, five-day-old seedlings grown on half-strength MS medium were transferred to MS medium supplemented with 0, 0.6, or 5 µM ABA, respectively. Primary root length was measured with ImageJ 1.48 (National Institutes of Health, Bethesda, Rockville Pike, MD, USA) after 10 days of treatment. Electrolyte leakage of ABA-treated plants was measured as described above.

Seven-day-old seedlings were transferred to 1/2 MS medium containing 125 mM NaCl supplemented without or with 5 µM ABA for 12 h. Seedlings were then incubated with H_2_DCF-DA (10 µM) for 5 min. Green fluorescence within cells was detected at the apical region of roots under a Leica confocal microscope. *AtRBOHD* and *AtRBOHF* transcription were detected with RT-qPCR. The Arabidopsis *AtACTIN2* was used as the internal control. Primers designed to target *AtRBOHD*, *AtRBOHF,* and *AtACTIN2* genes are shown in Appendix A.

### 4.14. The Expression of ABA-Mediated Salt Stress Responsive Genes

Wild-type, VC, and transgenic lines were grown on half-strength MS medium for five days. Then these seedlings were treated with 75 mM NaCl for 10 days on MS medium. Total RNA was extracted and used for the RT-qPCR assays (see above). The primers target to *ABF4, AREB2, ABI5, DREB1A, DREB2A, MYB2, RD29A, RAB18, SnRK2.2*, and *SnRK2.3* were listed in Appendix A.

### 4.15. Data Analysis

All experiments were repeated at least three times with consistent results. The data were subjected to SPSS (version 19.0, IBM Corporation, Armonk, NY, USA) for statistical analysis. Statistical analyses were performed using one–way ANOVA. Unless otherwise stated, a *p*-value less than 0.05 was considered statistically significant.

## 5. Conclusions

Salt treatment induced expression of a jacalin-related mannose-binding lectin, *JRL*, in *P. euphratica*. *PeJRL* overexpression contributed to salinity tolerance in Arabidopsis. A schematic model was proposed to show the mediation of PeJRL on ionic and ROS homeostasis (Figure 14). PeJRL upward-regulated activity of PM H^+^-pumps and Na^+^/H^+^ antiporters, thus promoted the Na^+^ extrusion. Furthermore, the activated H^+^-pumps preserved a less-depolarized membrane potential, which restrained the K^+^ loss through K^+^ channels in the PM [30]. In addition, the high-affinity K^+^ transporter, AtHKT1, might contribute to the avoidance of excessive accumulation of Na^+^ under salt stress. The *PeJRL* maintained ROS homeostasis by activating the antioxidant enzymes, and meanwhile repressing ABA biosynthesis, thus reduced the ABA-elicited ROS production and the oxidative damage during the period of salt stress.

## Figures and Tables

**Figure 1 ijms-20-00815-f001:**
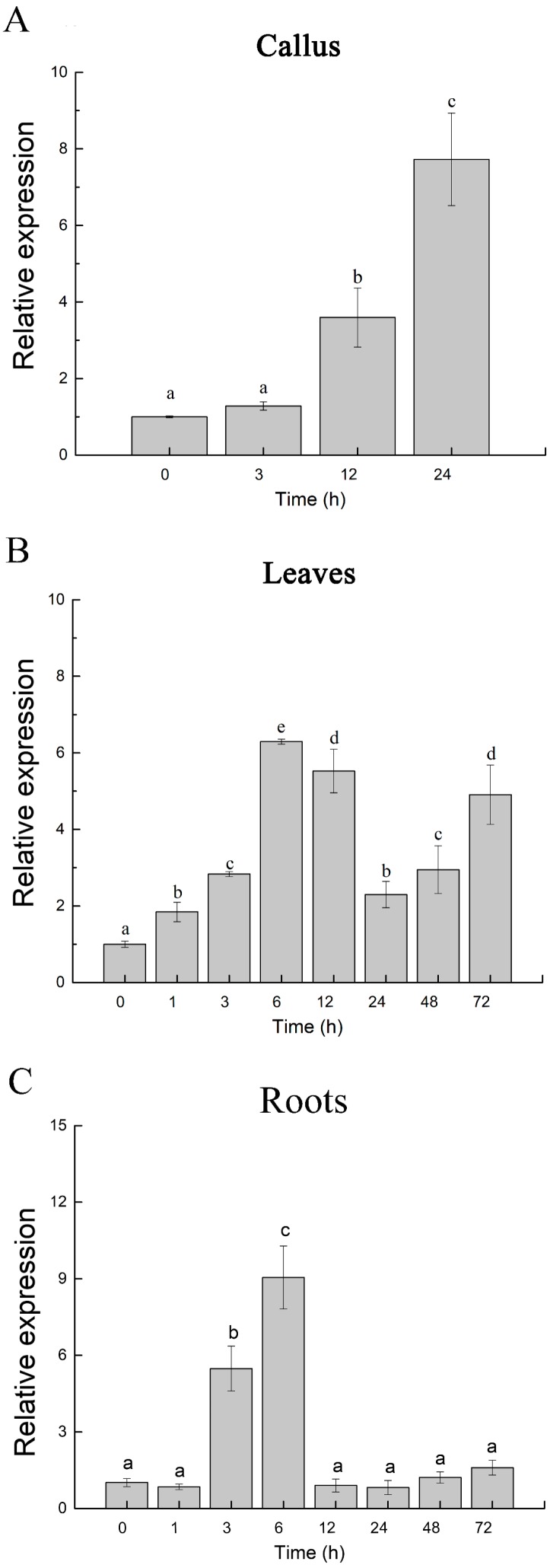
Expression level of jacalin-related mannose-binding lectin *PeJRL* gene in *Populus euphratica* callus cultures, leaves, and roots under NaCl treatment. (**A**) Callus cultures. *PeJRL* expression was measured using RT-qPCR during the period of salt treatment (125 mM NaCl, 24 h). (**B**) Leaves. (**C**) Roots. *PeJRL* expression was examined with RT-qPCR in young leaves and roots during NaCl stress (200 mM NaCl, 72 h). Expression levels of *PeJRL* in this experiment were calculated relative to the *P. euphratica* housekeeping gene, *PeACT7*, an internal reference. Primers designed to target *PeJRL* and *PeACT7* genes are shown in Appendix A. Each column corresponds to the mean of three independent replicates, and bars represent the standard error of the mean. Columns labelled with different letters (a–e) denote significant differences, at *p* < 0.05.

**Figure 2 ijms-20-00815-f002:**
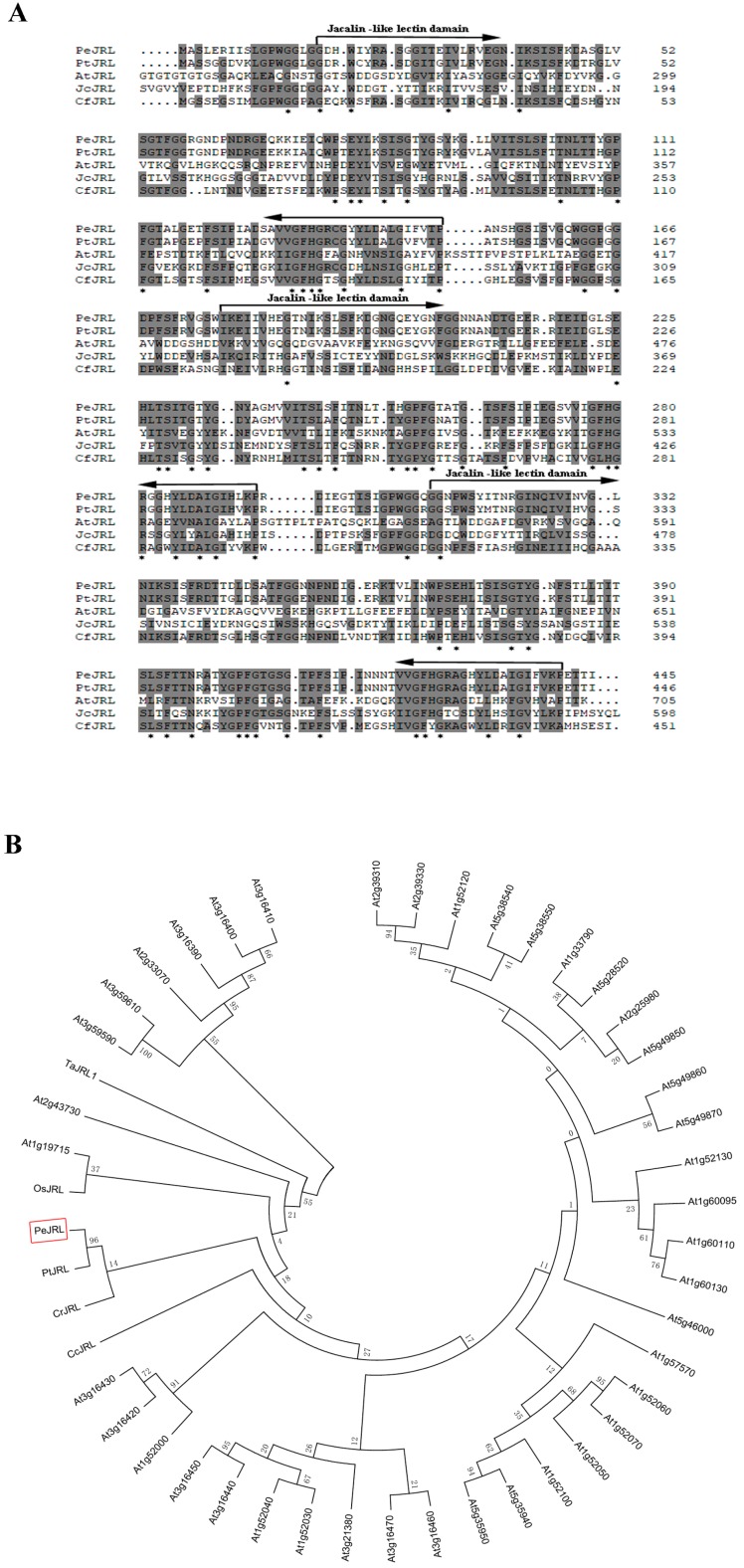
Comparative analysis of PeJRL proteins in different plant species. (**A**) Multiple sequence alignment of PeJRL with JRLs from other species. The JRL sequences are deduced from *Populus trichocarpa* (*PtJRL*, XP_024444915.1), *Arabidopsis thaliana* (*AtJRL*, AT3G16460), *Jatropha curcas* (*JcJRL*, XP_020533220.1), and *Cephalotus follicularis* (*CfJRL*, GAV61504.1). Grey shading and asterisks indicate conserved and invariant amino acid residues, respectively. Three jacalin-like lectin domains are shown with arrows. (**B**) Phylogenetic relationships between *PeJRL* and other *JRL* family members from different species. The phylogenetic tree was constructed with the neighbour-joining method using MEGA 5.2. Bootstrap values of 1000 replicates. At, *Arabidopsis thaliana*; Cc, *Castanea crenata*; Cr, *Cycas revolute*; Os, *Oryza sativa*; Pe, *Populus euphratica*; Pt, *Populus trichocarpa*; Ta, *Triticum aestivum.*

**Figure 3 ijms-20-00815-f003:**
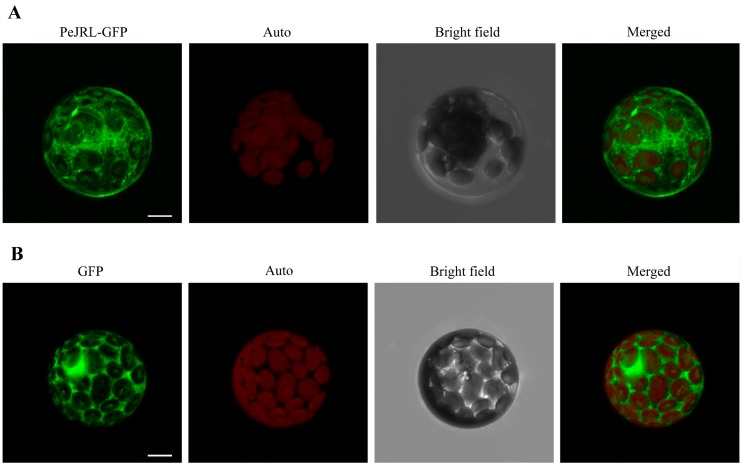
Subcellular localization of PeJRL by transient transformation in Arabidopsis mesophyll protoplasts. (**A**) Representative images of PeJRL-green fluorescent protein (GFP) in Arabidopsis protoplasts. (**B**) Control images for GFP vector. Green fluorescence was evenly detected in the cytosolic and nuclear spaces. Experiments were repeated four times with the same results. Scale bars = 5 μm.

**Figure 4 ijms-20-00815-f004:**
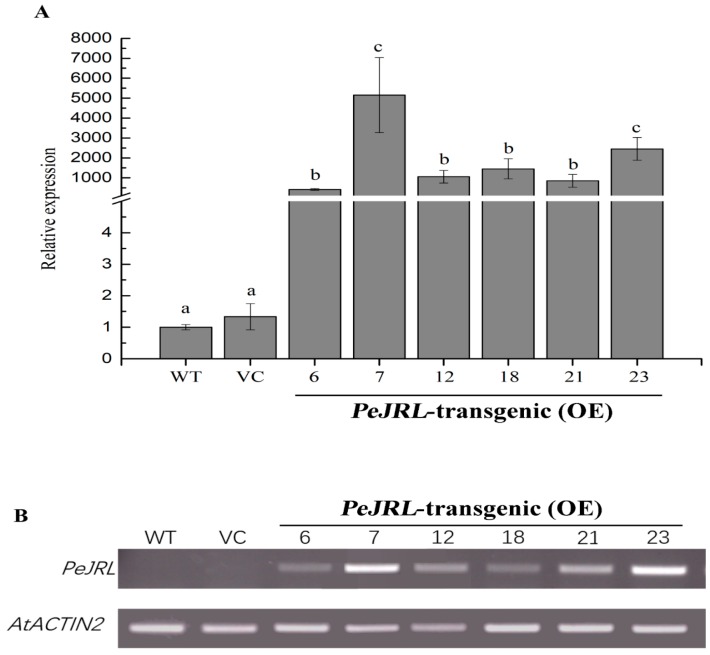
Overexpression of *PeJRL* in Arabidopsis. (**A**) RT-qPCR assays of *PeJRL* overexpression (OE) after transformation. *Arabidopsis thaliana* actin 2 (*AtACTIN2*) served as the internal control. Forward and reverse primers are shown in Appendix A. Each column corresponds to the mean of three individual plants, and bars represent the standard error of the mean. Columns labelled with different letters (a–c) indicate significant differences (*p* < 0.05) between wild-type (WT), vector control (VC), and transgenic lines. (**B**) Semi-quantitative reverse-transcription PCR (RT-PCR) analysis of *PeJRL* in WT, VC, and transgenic lines (#6, #7, #12, #18, #21, and #23).

**Figure 5 ijms-20-00815-f005:**
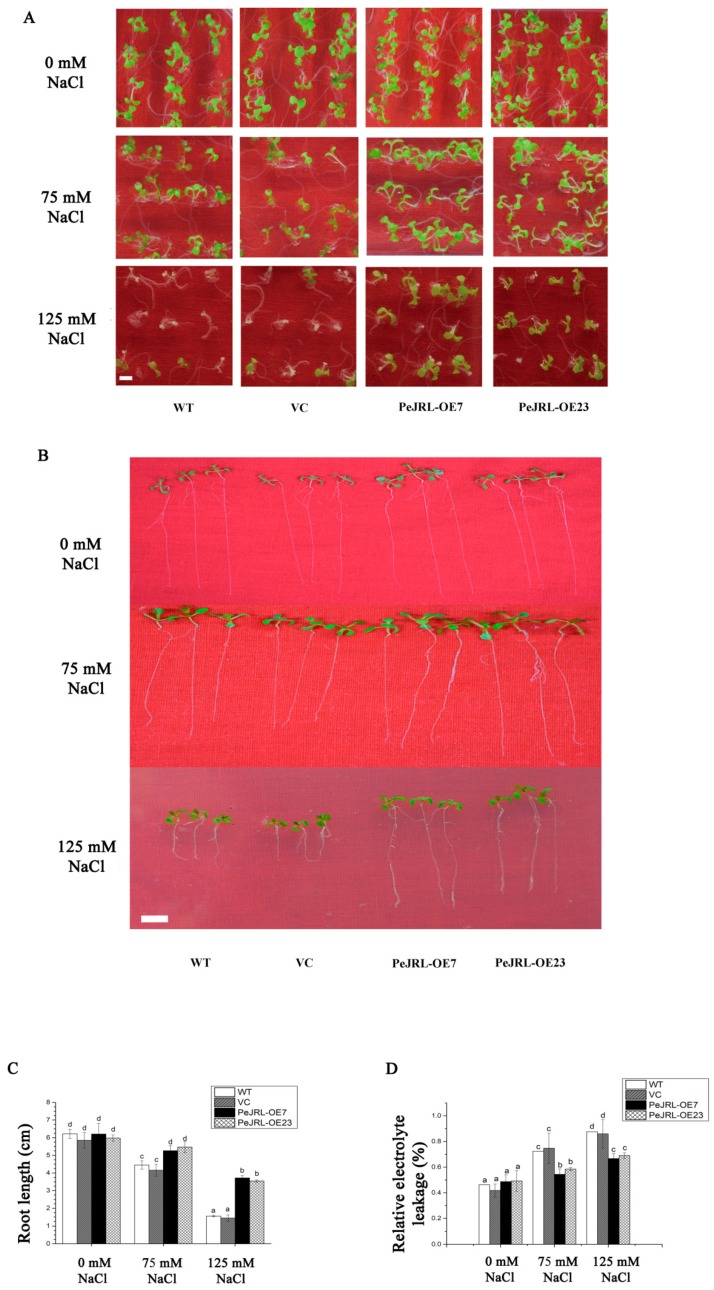
Salt tolerance tests in WT, VC, and *PeJRL*-transgenic Arabidopsis lines (OE7 and OE23). Seeds of WT, VC, and transgenic lines were allowed to germinate on 1/2 Murashige–Skoog (MS) medium supplemented with 0, 75 or 125 mM NaCl. Representative images show plant performance during seedling establishment (**A**) and root length (**B**,**C**) after 10 days of NaCl stress (0, 75 or 125 mM). Each column corresponds to the mean of three independent experiments (70 seeds for each test). (**D**) Relative electrolyte leakage. Electrolyte leakage was measured after 10 days of NaCl stress (0, 75 or 125 mM). Each column corresponds to the mean of three replicated experiments (20 seeds for each test) and bars represent the standard error of the mean. In **C** and **D**, columns labelled with different letters (a–d) denote significant differences at *p* < 0.05. Scale bar = 1 cm (**A**,**B**).

**Figure 6 ijms-20-00815-f006:**
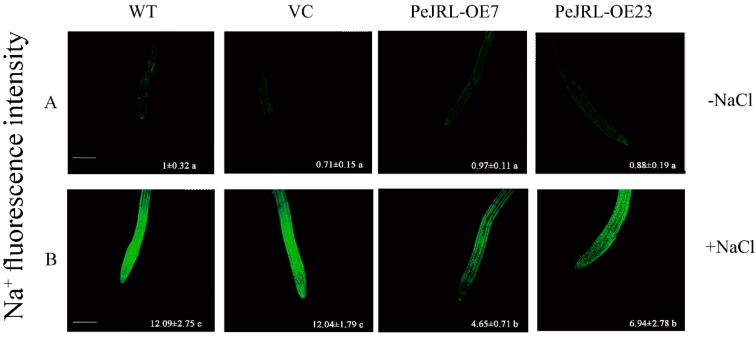
Na^+^ fluorescence intensity in root cells of wild-type (WT) Arabidopsis, vector control (VC), and *PeJRL*-transgenic lines (OE7 and OE23) under NaCl stress. Seven-day-old seedlings were exposed to 1/2 MS liquid medium supplemented with or without 125 mM NaCl for 12 h, then incubated with CoroNa-Green AM to detect Na^+^ concentrations. Green fluorescence in root cells was measured at the apical region of roots under a Leica confocal microscope. Representative confocal images show Na^+^ fluorescence intensity in control (−NaCl) (**A**) and salt-stressed roots (+NaCl) (**B**). Values (±SE) show the mean fluorescence intensity of CoroNa-Green AM. Each value corresponds to the mean of 6–9 independent plants. Values labelled with different letters (a–c) donate significant differences at *p* < 0.05. Scale bar = 250 µm.

**Figure 7 ijms-20-00815-f007:**
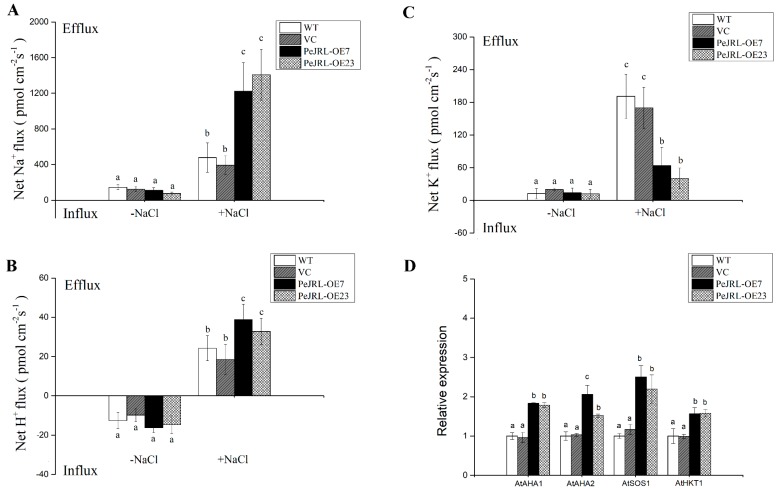
Effects of NaCl on Na^+^, K^+^, and H^+^ fluxes and transcription of K^+^/Na^+^ homeostasis genes in roots of wild-type (WT) Arabidopsis, vector control (VC), and *PeJRL*-transgenic lines (OE7 and OE23). Seven-day-old seedlings were treated without or with NaCl (125 mM) for 12 h. Steady-state flux profiles of Na^+^ (**A**), H^+^ (**B**), and K^+^ (**C**) in roots of WT Arabidopsis, VC, and *PeJRL*-transgenic lines (PeJRL-OE7 and PeJRL-OE23) were measured at the apical zone (ca 200 µm from the root apex) for 10 min. Mean fluxes of no-salt control (−NaCl) and salt-stressed roots (+NaCl) are shown. (**D**) Relative expression levels of *AHA1*, *AHA2*, *SOS1*, and *HKT1* in WT Arabidopsis, VC, and *PeJRL*-transgenic lines (OE7 and OE23) under NaCl treatment. Transcription of *AHA1*, *AHA2*, *SOS1*, and *HKT1* was detected with RT-qPCR. The Arabidopsis *AtACTIN2* was used as the internal control. Primers designed to target *AtAHA1*, *AtAHA2*, *AtSOS1*, and *AtHKT1*, and *AtACTIN2* genes are shown in Appendix A. Each column corresponds to the mean of three (**D**) to seven (**A**–**C**) independent plants, and bars represent the standard error of the mean. For each tested gene, columns labelled with different letters (a–c) denote significant difference between WT, VC, and transgenic lines at *p* < 0.05.

**Figure 8 ijms-20-00815-f008:**
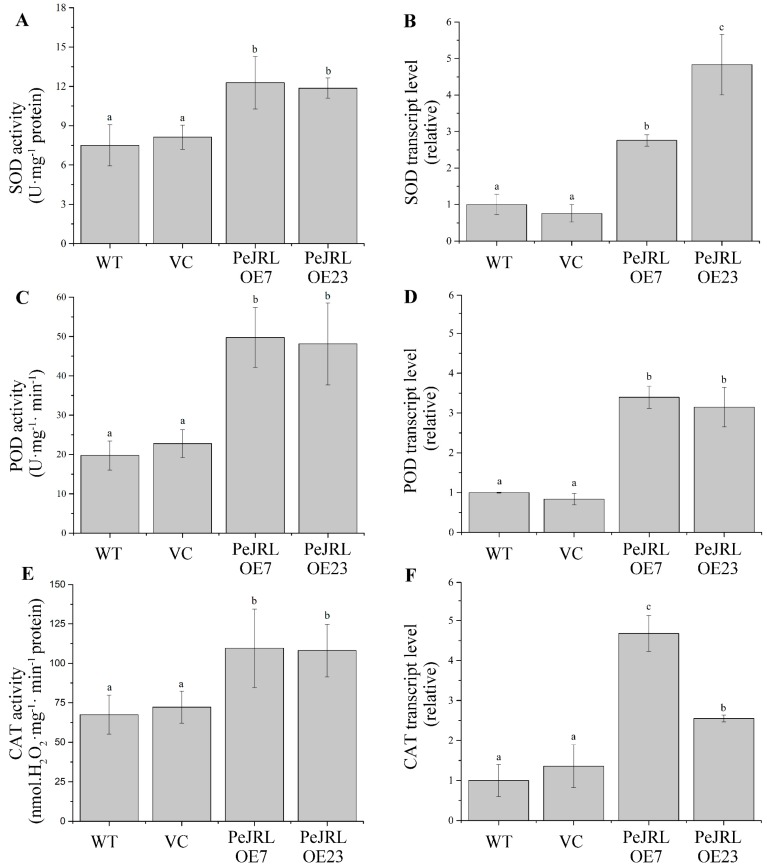
Effect of NaCl (75 mM, 10 d) on total activities of superoxide dismutase (SOD), catalase (CAT), peroxidase (POD), and transcription levels of encoding genes in wild-type (WT) Arabidopsis, vector control (VC), and *PeJRL*-transgenic lines (OE7 and OE23). (**A**,**C**,**E**) Total activities. (**B**,**D**,**F**) Transcription levels. Transcription of *AtSOD*, *AtPOD*, and *AtCAT* was detected with RT-qPCR. The Arabidopsis *AtACTIN2* was used as the internal control. Primers designed to target *AtSOD*, *AtPOD*, *AtCAT*, and *AtACTIN2* genes are shown in Appendix A. Each column corresponds to the mean of three replicated experiments, and bars represent the standard error of the mean. Columns labelled with different letters (a–c) denote significant difference at *p* < 0.05.

**Figure 9 ijms-20-00815-f009:**
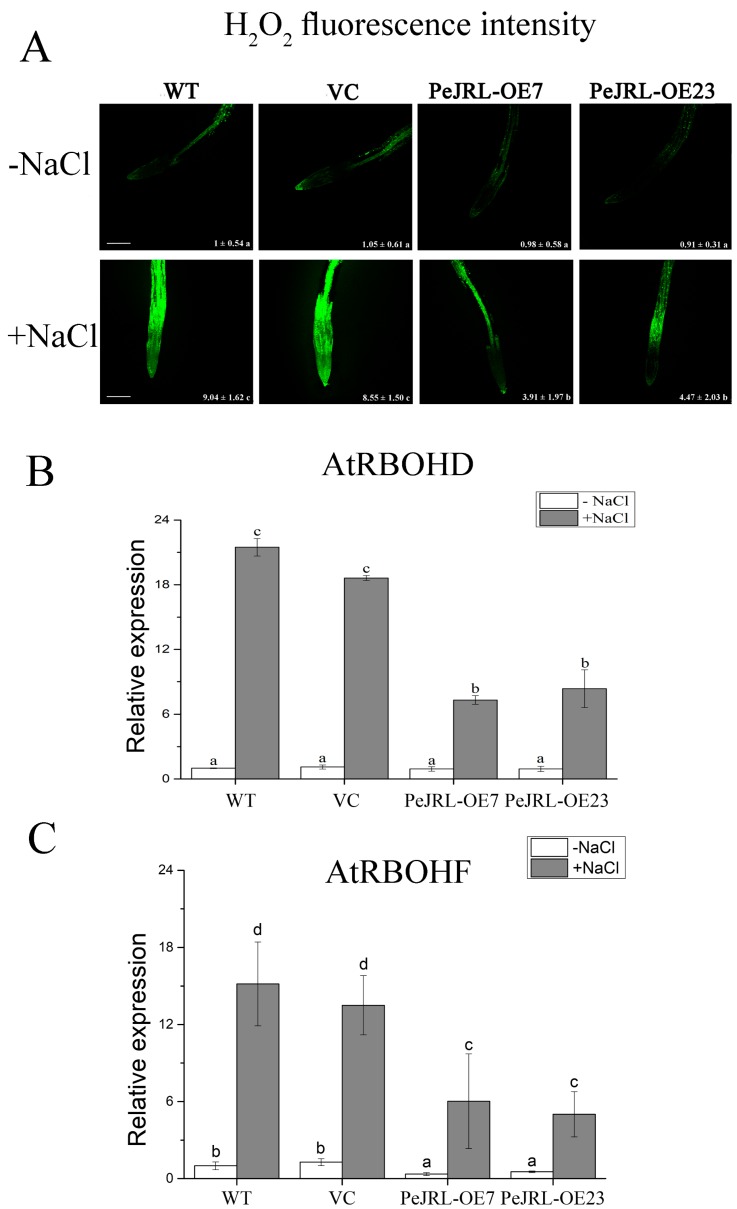
Effect of NaCl on root H_2_O_2_ production, *AtRBOHD* and *AtRBOHF* expression in wild-type (WT) Arabidopsis, vector control (VC), and *PeJRL*-transgenic lines (OE7 and OE23). (**A**) H_2_O_2_ fluorescence intensity. Seven-day-old seedlings were treated without or with NaCl (125 mM) for 12 h. Roots were then incubated with 2′, 7′-dichlorodihydro-fluorescein diacetate (H_2_DCF-DA, 10 µM) for 5 min. Green fluorescence within cells was detected at the apical region of roots under a Leica confocal microscope. Representative confocal images of H_2_O_2_ production in control (−NaCl) and salt-stressed roots (+NaCl) are shown. Values (±SE) show the mean fluorescence intensity of H_2_DCF-DA. Each value corresponds to the mean of three replicated experiments and values labelled with different letters (a–c) indicate significant differences at *p* < 0.05. (**B**) *AtRBOHD* and (**C**) *AtRBOHF* expression. Five-day-old seedlings were transferred to MS medium containing 0 or 75 mM NaCl for 10 days. *AtRBOHD* and *AtRBOHF* transcription were detected with RT-qPCR. The Arabidopsis *AtACTIN2* was used as the internal control. Primers designed to target *AtRBOHD*, *AtRBOHF*, and *AtACTIN2* genes are shown in Appendix A. Each column corresponds to the mean of seven independent plants, and bars represent the standard error of the mean. Columns labelled with different letters (a–d) denote significant difference at *p* < 0.05. Scale bar = 250 µm (**A**).

**Figure 10 ijms-20-00815-f010:**
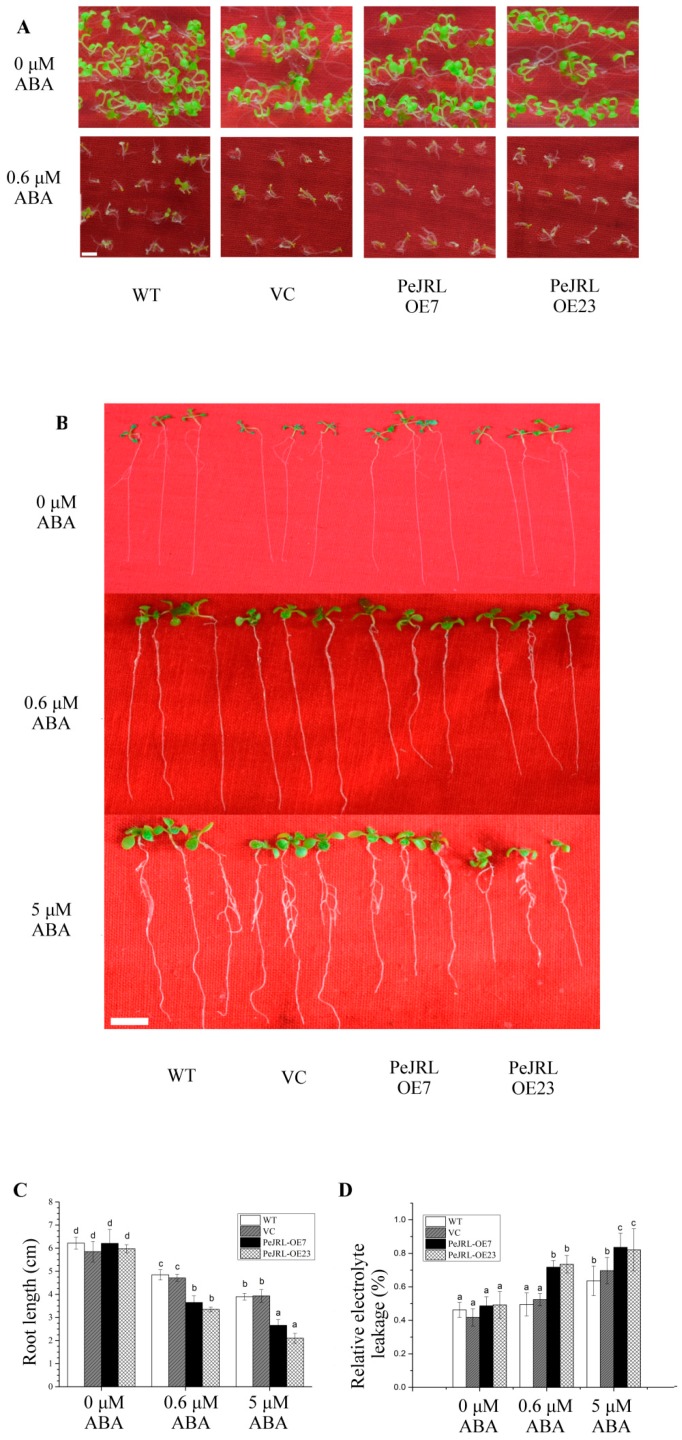
Abscisic acid (ABA) tests in wild-type (WT), vector control (VC), and *PeJRL*-transgenic Arabidopsis lines (OE7 and OE23). Seeds of WT, VC, and transgenic lines were allowed to germinate on MS medium supplemented with 0, 0.6 or 5 µM ABA. Representative images show plant performance during seedling establishment (**A**) and root length (**B**,**C**) after 10 days of ABA treatment (0, 0.6 or 5 µM). Each column in (**C**) corresponds to the mean of three independent experiments (70 seeds for each test), and bars represent the standard error of the mean. (**D**) Relative electrolyte leakage. Electrolyte leakage was measured after 10 days of ABA treatment (0, 0.6 or 5 µM). Each column corresponds to the mean of three replicated experiments (20 seeds for each test) and bars represent the standard error of the mean. In **C** and **D**, columns labelled with different letters (a–d) denote significant differences at *p* < 0.05. Scale bar = 1 cm (**A**,**B**).

**Figure 11 ijms-20-00815-f011:**
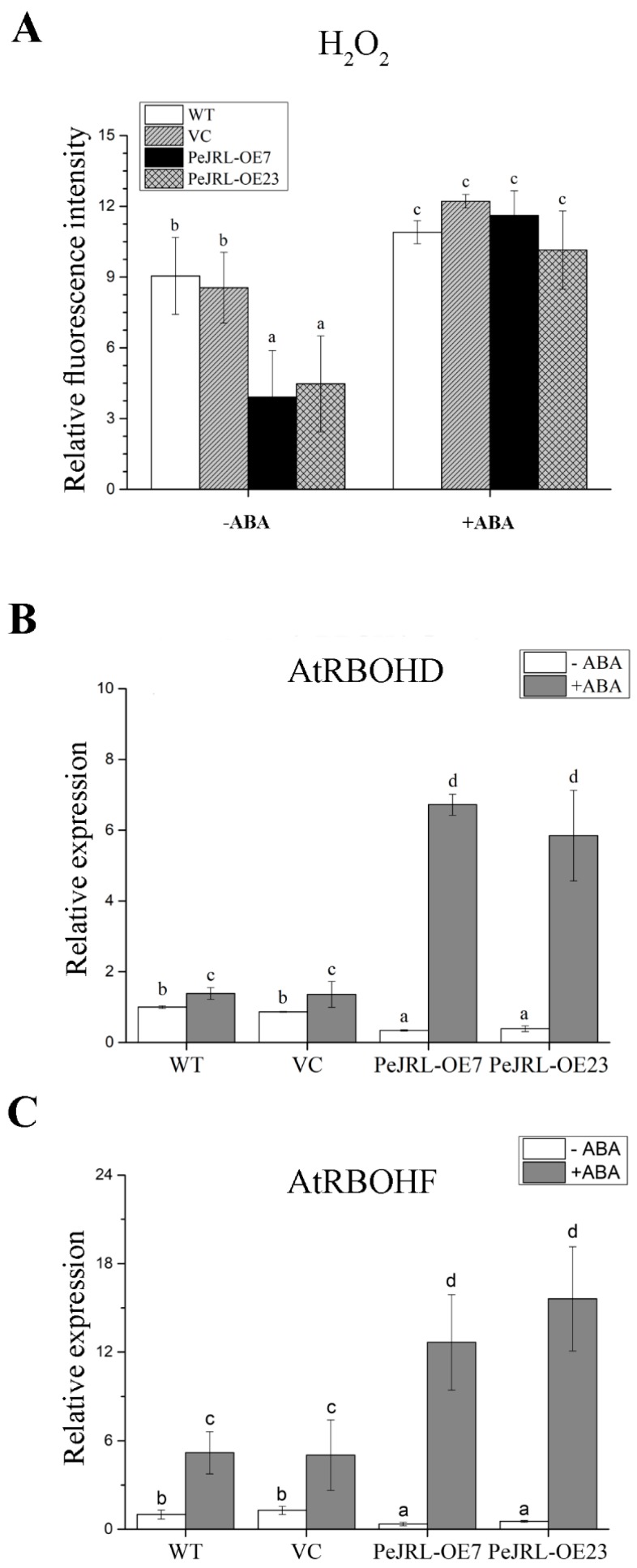
Effect of ABA (5 µM, 12 h) on root H_2_O_2_ production and *AtRBOHD* and *AtRBOHF* expression in wild-type (WT) Arabidopsis, vector control (VC), and *PeJRL*-transgenic lines (OE7 and OE23) under NaCl stress. (**A**) H_2_O_2_ production. Seven-day-old seedlings were transferred to 1/2 MS medium containing 125 mM NaCl supplemented without or with 5 µM ABA for 12 h. Seedlings were then incubated with H_2_DCF-DA for 5 min. Green fluorescence within cells was detected at the apical region of roots under a Leica confocal microscope. The mean fluorescence intensities of H_2_DCF-DA are shown in control (−ABA) and ABA-treated roots (+ABA). (**B**) *AtRBOHD* and (**C**) *AtRBOHF* expression. *AtRBOHD* and *AtRBOHF* transcription were detected with RT-qPCR. The Arabidopsis *AtACTIN2* was used as the internal control. Primers designed to target *AtRBOHD*, *AtRBOHF,* and *AtACTIN2* genes are shown in Appendix A. Each column corresponds to the mean of seven independent seedlings and columns labelled with different letters (a–d) indicate significant differences at *p* < 0.05.

**Figure 12 ijms-20-00815-f012:**
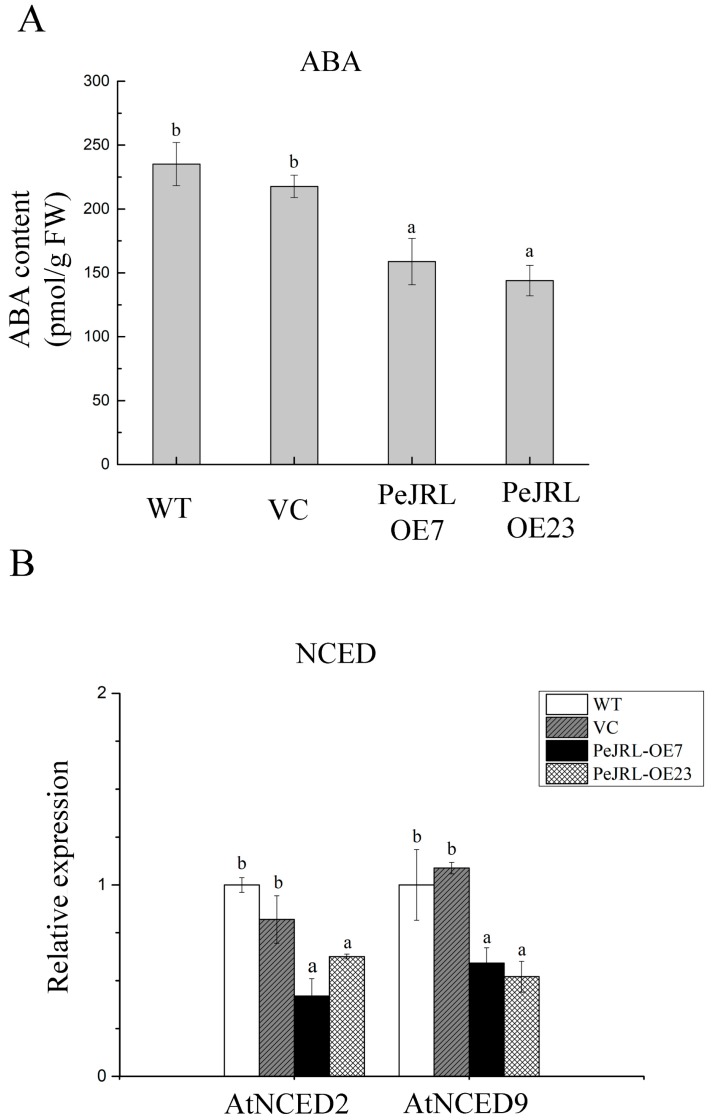
ABA content and expression of *AtNCED2* and *AtNCED9* in roots of wild-type (WT) Arabidopsis, vector control (VC), and *PeJRL*-transgenic lines (OE7 and OE23) under NaCl stress. Five-day-old seedlings were transferred to MS medium containing 75 mM NaCl for 10 days. (**A**) ABA content. ABA concentration was estimated using the enzyme-linked immunosorbent assay (ELISA) test. (**B**) Expression of *AtNCED2* and *AtNCED9*. *AtNCED2* and *AtNCED9* transcription were detected with RT-qPCR. The Arabidopsis *AtACTIN2* was used as the internal control. Primers designed to target *AtNCED2*, *AtNCED9*, and *AtACTIN2* genes are shown in Appendix A. Each column corresponds to the mean of seven independent seedlings and columns labelled with different letters (a, b) indicate significant differences at *p* < 0.05.

**Figure 13 ijms-20-00815-f013:**
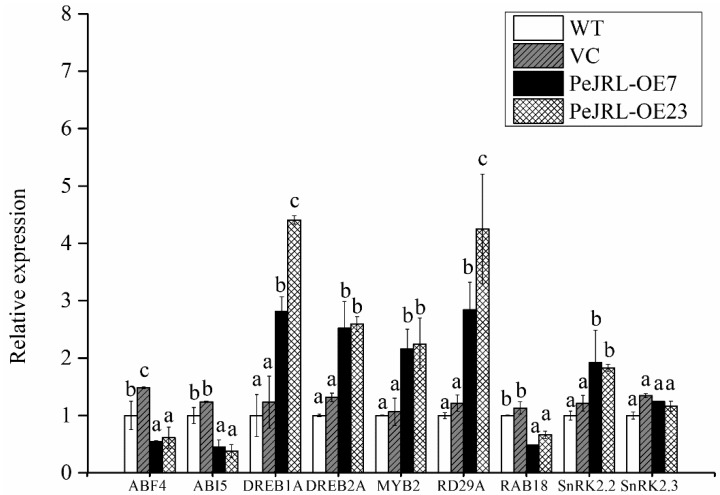
Expression level of ABA-mediated salt stress responsive genes in wild-type (WT) Arabidopsis, vector control (VC), and *PeJRL*-transgenic lines (OE7 and OE23) under salt stress. Five-day-old seedlings were transferred to MS medium containing 75 mM NaCl for 10 days. Transcription of ABA-mediated salt stress responsive genes was detected with RT-qPCR. The Arabidopsis *AtACTIN2* was used as the internal control. Primers designed to target *AtACTIN2* and ABA-mediated salt stress responsive genes are shown in Appendix A. For each tested gene, each column corresponds to the mean of three independent replicates and columns labelled with different letters (a–c) indicate significant differences between WT, VC, and transgenic lines at *p* < 0.05.

**Figure 14 ijms-20-00815-f014:**
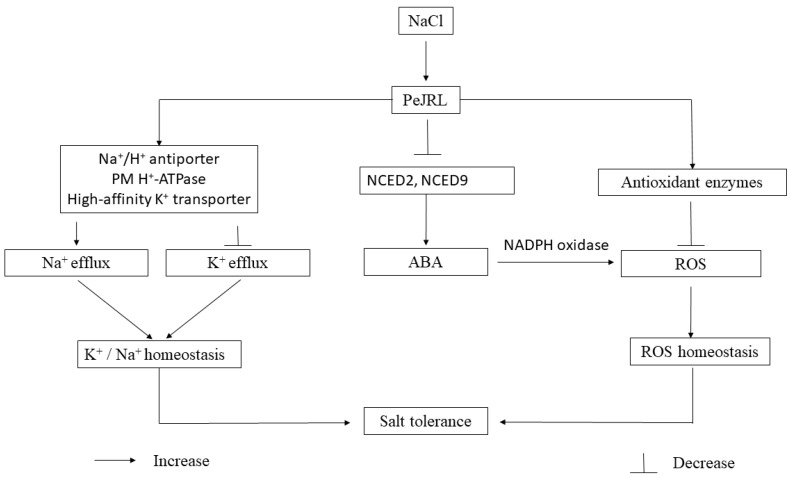
A schematic model showing mediation of *Populus euphratica* JRL in plant response to salt stress.

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
