# Peer review of "Populus euphratica* JRL Mediates ABA Response, Ionic and ROS Homeostasis in Arabidopsis under Salt Stress"

_ijms, 2019, doi:10.3390/ijms20040815_

Round 1
Reviewer 1 Report
The present manuscript by Huilong Zhang et al. provides a comprehensive study on the role of jacalin-related mannose binding lectin PeJRL from salt-resistant poplar in plant response to salinity stress. Ectopic expression of PeJRL in Arabidopsis improved seedling growth in the presence of 125 mM NaCl in comparison to the wild-types. The authors characterized the phenomenon of enhanced salt tolerance of transgenic Arabidopsis showing improved capacity to maintain ionic homeostasis and effective scavenging of reactive oxygen species. On the contrary, overexpression of PeJRL in Arabidopsis negatively affected ABA metabolism causing lower ABA levels, enhanced ABA sensitivity of seedlings and significantly increased expression of NADPH upon salt treatment in the presence of ABA that was not observed in wild type plants. Results presented in this manuscript showed that PeJRL modulate cellular stress response at different levels and contribute significantly to salt stress tolerance in Arabidopsis.
But there are some concerns, as detailed below
The interpretation of cellular localization of PeJRL is not accurate. It seems that PeJRL is predominantly cytoplasmic. The quality of the images in Figure 3B should be better because FM4-64 staining is easy to perform and enables visualization of the cell outline. Even if the pool of GFP-tagged protein that associates with plasma-membrane is small it has quite characteristic appearance, it completely outlines the shape of the cell and highlights cell border. To convince the reader that PeJRL associated with PM I suggest replacing the images otherwise the association with PM is rather doubtful until there are no other evidences that might confirm this localization.
Lack of comment concerning nuclear localization of PeJRL although there is a high similarity between PeJRL and orthologous lectins with nucleocytoplasmic distribution.
Please improved the Figures quality:
Figure 1- please replace the “relatived expression” to “relative expression”
Figure 2 – larger font size should be used ,
Figure 3 – lack of scale bars
Figure 4 – the abbreviations VC is not explained although it is used for the first time in figure description
Figure 5, 8, 10, 11, 12 - larger and equal font size should be used for image and graph labels,
Figure 11 please replace “relative intensity” to “relative fluorescence intensity”
Keywords: the abbreviations NMT is not explained
P.7 line 11 the abbreviation NMT is used for the first time in the text and is not explained
P. 14 line 6 – Figure 3 shows that PeJRL is mainly cytoplasmic, this information should be emphasized in the text, association with PM is rather putative
P. 15 line 30 – the sentence is true but only for transgenic plants, it cannot be a general statement since the expression of NADPH oxidase is not affected by combined treatment with ABA and NaCl in wild-type plants (Figure 11)
P. 16 line – please check the settings for FM4-64 confocal imaging, the manufacturer recommend ~ 515 nm excitation wave length and if colocalization with GFP is performed it is not recommended to excite with the same laser wave length different fluorescent tags
P. 18 line 6 – in the whole text there is lack of information that the ionic fluxes were measured with the use of selective microelectrode, please add the information here
Author Response
Authors response to Reviewer 1
Comments and Suggestions for Authors
The present manuscript by Huilong Zhang et al. provides a comprehensive study on the role of jacalin-related mannose binding lectin PeJRL from salt-resistant poplar in plant response to salinity stress. Ectopic expression of PeJRL in Arabidopsis improved seedling growth in the presence of 125 mM NaCl in comparison to the wild-types. The authors characterized the phenomenon of enhanced salt tolerance of transgenic Arabidopsis showing improved capacity to maintain ionic homeostasis and effective scavenging of reactive oxygen species. On the contrary, overexpression of PeJRL in Arabidopsis negatively affected ABA metabolism causing lower ABA levels, enhanced ABA sensitivity of seedlings and significantly increased expression of NADPH upon salt treatment in the presence of ABA that was not observed in wild type plants. Results presented in this manuscript showed that PeJRL modulate cellular stress response at different levels and contribute significantly to salt stress tolerance in Arabidopsis.
Authors: The authors would like to thank the peer Reviewers very much for the careful
review on this manuscript. Substantial revisions were made according to your valuable comments. Indicated changes were labelled with red in the text. Please see below the point-to-point response to Reviewers’ comments and queries.
Reviewer 1: 1.
a. The interpretation of cellular localization of PeJRL is not accurate. It seems that PeJRL is predominantly cytoplasmic. The quality of the images in Figure 3B should be better because FM4-64 staining is easy to perform and enables visualization of the cell outline. Even if the pool of GFP-tagged protein that associates with plasma-membrane is small it has quite characteristic appearance, it completely outlines the shape of the cell and highlights cell border. To convince the reader that PeJRL associated with PM, I suggest replacing the images otherwise the association with PM is rather doubtful until there are no other evidences that might confirm this localization.
b. Lack of comment concerning nuclear localization of PeJRL although there is a high similarity between PeJRL and orthologous lectins with nucleocytoplasmic
distribution.
Authors: According to your valuable suggestions, the original Figure 3B was removed because the visualization of the cell outline labelled by FM4-64 is not a convinced colocalization of plasma membrane. The related changes were also made in Results (Lines 141-145), Discussion (Lines 360-363), and Materials and Methods (Lines 461-470).
In addition, the comment concerning nuclear localization of PeJRL was also addressed in the Discussion section (Lines 360-363).
Reviewer 1: 2. Please improve the Figures quality:
Figure 1- please replace the “relatived expression” to “relative expression”
Figure 2 – larger font size should be used,
Figure 3 – lack of scale bars
Figure 4 – the abbreviations VC is not explained although it is used for the first time in figure description
Figure 5, 8, 10, 11, 12 - larger and equal font size should be used for image and graph labels,
Figure 11 please replace “relative intensity” to “relative fluorescence intensity” Authors: We made indicated changes in the Figures to improve the quality, in brief,
Figure 1- the “relatived expression” was replaced with “relative expression”;
Figure 2- larger font size was used;
Figure 3- scale bars were added;
Figure 4- the abbreviations of VC, WT, OE were explained in the figure legends; Figure 5, 8, 10, 11, 12 - larger and equal font size were used for image and graph
labels;
Figure 11- “relative intensity” was changed to “relative fluorescence intensity”.
Reviewer 1: 3. Keywords: the abbreviations NMT is not explained
Authors: As you suggested, the abbreviation of NMT was addressed (Line 37).
Reviewer 1: 4. P7 line 11 the abbreviation NMT is used for the first time in the text and is
not explained
Authors: Indicated change was made in revised manuscript (Lines 208-211).
Reviewer 1: 5. P14 line 6 – Figure 3 shows that PeJRL is mainly cytoplasmic, this information should be emphasized in the text, association with PM is rather putative.
Authors: As suggested, the cytoplasmic PeJRL was emphasized in the text (Lines 141-145). The original Figure 3B showing the putative colocalization with plasma membrane was removed from the revised manuscript.
Reviewer 1: 6. P15 line 30 – the sentence is true but only for transgenic plants, it cannot be a general statement since the expression of NADPH oxidase is not affected by combined treatment with ABA and NaCl in wild-type plants (Figure 11). Authors: Thank you very much for your suggestion. The statement was changed to “Under NaCl salinity, ABA-induced increase of H2O2 was more pronounced in PeJRL-overexpressed plants (2.9 and 2.3-fold), as compared to WT and VC (1.2 and
1.4-fold) (Figure 11). This is consistent to the drastic rise of AtRBOHD and AtRBOHF in salinized transgenic plants; while the expression of NADPH oxidase was less affected by combined treatment with ABA and NaCl in wild-type plants (Figure 11)” (Lines 420-425)
Reviewer 1: 7. P16 line – please check the settings for FM4-64 confocal imaging, the manufacturer recommend ~ 515 nm excitation wave length and if colocalization with GFP is performed it is not recommended to excite with the same laser wave length
different fluorescent tags.
Authors: Thanks a lot for your valuable suggestion. Indeed it is not recommended to excite with the same laser wave length for FM4-64 and GFP. In our revised manuscript, the colocalization results related with FM4-64 was removed based on your comments. (Lines 462-470)
Reviewer 1: 8. P18 line 6 – in the whole text there is lack of information that the ionic fluxes were measured with the use of selective microelectrode, please add the information here.
Authors: As suggested, the information about the using of selective microelectrode for ionic fluxes recording was addressed in the revision. (Lines 539-550)
Authors: Finally, the authors would like to thank the reviewer again for the valuable time, careful editing, and helpful suggestions. We wish that the revision is sufficient to
satisfy your positive considerations. In addition, we also made indicated changes according review comments from Reviewer 2 and 3.
Reviewer 2 Report
It is a well-written and well-organized manuscript. However, there are several questions or comments to understand this manuscript.
What about adding the explanation about salt stress (abiotic stress) in plant at the introduction?
What about showing protein level of PeJRL? Although RT-PCR results indicated that OE7 and OE23 are over-expressed, protein abundance would be more acceptable.
Most of Arabidopsis phenotype related with salt stress responses have been presented in the root. What about showing gene expression of PeJRL in the different tissue such as root, as performed in the leaves (fig 1)? Does PeJRL is leaf-specific gene?
What’s difference between Fig 9B(-NaCl) and Fig 10B (-ABA)?
Authors described all results at each section. However, it’s difficult to figure out what’s the conclusive result and what authors are suggesting thing through the result. Please, add more explanation why authors designed those experiments and what authors suggest. For example, what’s the interpreted meaning of the subcellular localization of PeJRL (2.3 section)?
What about showing ABA-mediated salt stress responsive genes such as DREBs, RD29?
What’s the expression value of RBOHF gene in the transgenic plants either NaCl (ABA) treatment or not?
Author Response
Authors response to Reviewer 2
It is a well-written and well-organized manuscript. However, there are several questions or comments to understand this manuscript.
Authors: The authors would like to thank the peer Reviewers very much for the careful review on this manuscript. Substantial revisions were made according to your valuable comments. Indicated changes were labelled with red in the text. Please see below the point-to-point response to Reviewers’ comments and queries.
Reviewer 2: 1. What about adding the explanation about salt stress (abiotic stress) in plant at the introduction?
Authors: As suggested by the Reviewer, the adverse of salt stress and plant response to salinity were added to the Introduction. (Lines 40-46).
Reviewer 2: 2. What about showing protein level of PeJRL? Although RT-PCR results indicated that OE7 and OE23 are over-expressed, protein abundance would be more acceptable.
Authors: The actual protein level would be better to indicate the expression of PeJRL in OE7 and OE23, as compared to RT-PCR results. We have previously shown that the overexpression of Populus euphratica APYRASE2 gene resulted in an enhanced abundance in Arabidopsis [1]. Therefore, the increase in mRNA levels is generally related to protein levels [1]. PeJRL protein level was not detected in this study. Our phenotype tests clearly showed that transgenic plants overexpressing PeJRL resulted in an enhanced tolerance to salinity, suggesting that PeJRL plays an important role in the adaptation to saline environment.
Reviewer 2: 3. Most of Arabidopsis phenotype related with salt stress responses have been presented in the root. What about showing gene expression of PeJRL in the different tissue such as root, as performed in the leaves (Fig 1)? Does PeJRL is leaf-specific gene?
Authors: We carried out supplemental experiment to testify the gene expression of PeJRL in P. euphratica roots. Total RNA was extracted from roots, followed by reverse transcription to obtain cDNA. RT-qPCR showed that PeJRL transcript level increased rapidly at 3 h after salt treatment, reaching the peaking at 12 h, and then returned to pretreatment level (Figure 1C). Our data showed that PeJRL is not a leaf-specific gene (Lines 100-106).
Reviewer 2: 4. What’s difference between Fig 9B(-NaCl) and Fig 10B (-ABA)?
Authors: Fig 9B(-NaCl) indicated no-salt control plants, while Fig 11B (-ABA) showed NaCl-treated plants but without ABA treatment. The different treatments were explained in the figure legends of Fig 9 and Fig 11.
Reviewer 2: 5. Authors described all results at each section. However, it’s difficult to figure out what’s the conclusive result and what authors are suggesting thing through the result. Please, add more explanation why authors designed those experiments and what authors suggest. For example, what’s the interpreted meaning of the subcellular localization of PeJRL (2.3 section)?
Authors: Thank you very much for your valuable comments. As you suggested, more explanation of experiment design and the aim were addressed in the Result section. All revisions were labelled with red in the text.
Reviewer 2: 6. What about showing ABA-mediated salt stress responsive genes such as DREBs, RD29?
Authors: As suggested by the Reviewer, a subset of ABA-mediated salt stress responsive genes were screened in all tested lines and shown in Figure 13. These genes include ABF4/AREB2 [2], ABI5 [3], DREB1A, DREB2A [4], MYB2 [5], RD29A [6], RAB18[7], SnRK2.2, and SnRK2.3 [8,9]. RT-PCR data showed that the PeJRL-overexpressed lines showed higher transcript levels of DREB1A, DREB2A, MYB2, RD29A, and SNRK2.2 compared with those of WT plants. The up-regulated ABA signaling network might explain the ABA sensitivity in transgenic plants (Lines 336-345).
Reviewer 2: 7. What’s the expression value of RBOHF gene in the transgenic plants either NaCl (ABA) treatment or not?
Authors: According to your valuable advice, in our supplemental experiments the expression of RBOHF was measured in parallel with RBOHD gene in wild-type, VC and transgenic lines. The experimental data was addressed in Figure 9 (NaCl) and in Figure 11 (ABA) respectively to show the gene expression upon NaCl or ABA treatment.
Authors: Finally, the authors would like to thank the reviewer again for the valuable time, careful editing, and helpful suggestions. We wish that the revision is sufficient to satisfy your positive considerations. In addition, we also made indicated changes according review comments from Reviewer 1 and 3.
Additional References
1. Deng, S. R.; Sun, J.; Zhao, R., Ding, M. Q.; Zhang, Y. N.; Sun, Y.l.; Wang, W.; Tan, Y. Q.; Liu, D. D.; Ma, X. J.; Hou, P. C.; Wang, M. J.; Lu, C. F.; Shen, X.; Chen, S. L. Populus euphratica APYRASE2 Enhances Cold Tolerance by Modulating Vesicular Trafficking and Extracellular ATP in Arabidopsis Plants. Plant Physiol. 2015, 169, 530-548.
2. Choi, H.; Hong, J.; Ha, J.; Kang, J.; Kim, S. Y. ABFs, a family of ABA-responsive element binding factors. Journal of Biological Chemistry. 2000, 275, (3), 1723-30.
3. Finkelstein, R. R.; Lynch, T. J. The Arabidopsis Abscisic Acid Response Gene ABI5 Encodes a Basic Leucine Zipper Transcription Factor. Plant Cell. 2000, 12, (4), 599.
4. Liu, Q.; Kasuga, M.; Sakuma, Y.; Abe, H.; Miura, S.; Yamaguchi-Shinozaki K.; Shinozaki, K. Two transcription factors, DREB1 and DREB2, with an EREBP/AP2 DNA binding domain separate two cellular signal transduction pathways in drought-and low-temperature-responsive gene expression, respectively, in Arabidopsis. Plant Cell. 1998, 10, (8), 1391-406.
5. Abe, H.; Urao, T.; Ito, T.; Seki, M.; Shinozaki, K.; Yamaguchishinozaki, K. Arabidopsis AtMYC2 (bHLH) and AtMYB2 (MYB) function as transcriptional activators in abscisic acid signaling. Plant Cell. 2003, 15, (1), 63.
6. Yamaguchishinozaki, K.; Shinozaki, K. A novel cis-acting element in an Arabidopsis gene is involved in responsiveness to drought, low-temperature, or high-salt stress. Plant Cell. 1994, 6, (2), 251-264.
7. Lång, V.; Palva, E. T. The expression of a rab-related gene, rab18, is induced by abscisic acid during the cold acclimation process of Arabidopsis thaliana (L.) Heynh. Plant molecular biology. 1992, 20, (5), 951-962.
8. Fujii, H.; Chinnusamy, V.; Rodrigues, A.; Rubio, S.; Antoni, R.; Park, S. Y.; Cutler, S. R.; Sheen, J.; Rodriguez, P. L.; Zhu, J. K. In vitro reconstitution of an abscisic acid signalling pathway. Nature. 2009, 462, (7273), 660.
9. Fujii, H.; Verslues, P. E.; Zhu, J. K. Identification of two protein kinases required for abscisic acid regulation of seed germination, root growth, and gene expression in Arabidopsis. Plant Cell. 2007, 19, (2), 485-94.
Reviewer 3 Report
The manuscript of Zhang and collaborators characterizes JRL gene from Populus euphratica. Their goal is to investigate the role of PeJRL in salinity tolerance, expressing the Populus gene in Arabidopsis. They propose a model where salt treatment induces the expression of this lectin that mediates regulation on ion and ROS homeostasis and represses ABA regulated genes to improve salt tolerance.
This manuscript follows a molecular and genetic approach, all the experiments show basic analysis nonetheless the data reported gives a clear message, PeJRL is involved in abiotic stress, specifically in salt stress.
The introduction is well organized and contributes to the basic knowledge to understand the results section. The experiments are well designed and executed, the amount of negative controls is appropriate and the content in the pictures is suitable, but it would be necessary to polish some of the figures. The discussion and materials/methods sections are correct even some mistakes and spelling should be addressed.
The general overview of the manuscript is positive, but cohesion is required among different parts of the manuscript as well as proofreading of the text.
General comments:
The results section of the manuscript lacks cohesion and coherence. I miss an introductory sentence in each paragraph that focuses the aim of the experiment and a good linkage between experiments. For example, after section 2.5, the overexpression of the PeJRL, the authors do not introduce the goal of the next experiments. This is happening in each of the remaining paragraphs.
The manuscript needs a very careful review of all the details in the experiments, how the figures displayed, and the nomenclature used (acronyms, abbreviations, etc.). The authors should explain the acronym/abbreviation the first time they mention it and then continue using the abbreviation. They must be consistent about it along the manuscript. For example, they use WT and wild-type indistinctly.
The materials and methods section should be reviewed in depth. There are mistakes showing that this manuscript has not been revised properly before submission.
Minor concerns:
Page 2
Line40. In section 2.1, the authors comment the expression of JRL in P. euphratica leaves but they do not explain or give any idea regarding the big drop of expression showed at 24h and why they think the plant increases the transcript production afterward.
Page 3
Figure2. I found this figure very small and is not easy to visualize.
Page 4
Figure 3. The authors forgot the scale bar.
Line 24. The title numeration is wrong. It should be the section 2.4 instead of 2.5.
Page 5
Line 6. The authors forgot to explain the acronyms WT and VC. They explain them later in the manuscript but that should be done the first time they mention them which is here. They should be consistent in the nomenclature.
Line 9. The same problem with the section numeration, this should be section 2.5.
Line 15. Authors start this paragraph mentioning electrolyte leakage but without explaining the reason they are performing that analysis. I would suggest a brief introduction.
Page 6
Figure 5A and C. Very unfortunate choice of background. The red color in figure 5C does not allow the correct visualization of Arabidopsis roots. I will encourage them to try to supply more contrasted pictures and for future experiments, I will suggest them to use a black background.
Figure 5A. The scale bar is very difficult to see, too thin.
Line 5. “survival rate”. This is not a survival rate experiment, the picture is showing seedling establishment after 10 days growing in NaCl supplemented plates. Please correct the misunderstanding appropriately in the remaining text and figures.
Line 8. “Each column is…” I would suggest the authors rephrasing it, for example, “Each column corresponds to the mean of…”
Line 12. Same problem with section numeration.
Line 13. I would suggest the authors explain briefly when they use reagents, such as “CoroNa Green”, in this paragraph.
Line 15 and 16. The authors mention figure 6A and B but there are no such letters in figure 6.
Page 7
Figure 6. I suggest the authors explain the left side of the figure more. “Na+” is not enough. Are they measuring levels? Which units?
Line 10. Same problem with section numeration.
Line 11. “NMT”, the acronym description is missing.
Line 14. “were notably higher”, a percentage would help here.
Lines 17-20. The authors did not connect this paragraph with the one before. Can the authors link the fluxes in the root with their decision to measure gene expression?
Page 8
Line 15. Same problem with section numeration.
Line 15. “Activity and transcript...” what do the authors mean with that?
Line 16. Introduction to why they decide to measure the activities of anti-oxidative enzymes is missing.
Page 9
Line 10. Same problem with section numeration.
Line 11. Why do the authors decide to analyze H2O2? Explain in the paragraph.
Page 10
Figure 9A. I would suggest the authors explain the title of the figure more. “H2O2” is not enough. Are they measuring levels? Which units?
Line 16. Same problem with section numeration.
Line 17. The authors do not explain why they decided to study ABA in this section.
Page 11
Figure 10. Background problem again.
Figure 10. The letter size in the figure is different in each graph, I would suggest adopting one size all over the figure, if it is possible, to keep the balance.
Page 14
Line 25. “… transgene plants”. Do the authors mean transgenic plants?
Page 15
Line 1. “… enabled transgenic plants to maintain K+ under saline conditions…” What do the authors mean with K+? Is it K+ levels?
Line 10. “… exposure in transgene plants”. They are transgenic plants.
Page 16
Line 25. Explain acronym “PM-specific probe”.
Line 30. Correct the nomenclature of the vector “PMDC85-the” to pMDC85 and the same in line 31 and 33.
Line 33. Correct “(Clo-0)” twice in this line.
Line 41 and 42. Modify sentence to, “semi-quantitative reverse-transcription PCR (RT-PCR)” and use appropriately in the remaining paragraph.
Line 44. Rephrase this: “DNA in isolated RNA was eliminated with RNase-free DNase…”
Page 17
Line 10. I suggest them to use MS medium supplemented with instead of “MS medium with the addition of”.
Line 12. “under light conditions…” do the authors mean, continuous light? Or long day, short day conditions? Please specify.
Line 14. What do the authors mean with “the seedlings were salinized”? The seedlings were treated with…
Line 42. “DCF” Is this the same reagent that in line 37? Why is now called differently?
Page 18
Line 1. Transgene plants again.
Line 14. “…enzyme assays”, enzymatic assays
Line 31. “After 3 days of incubation at 4°C”, this process is called seed stratification.
Line 32. “under light conditions” correct as previously suggested.
Line 32. Correct “survival rate” as mentioned before. This is a seed germination experiment and the measurement at 10 days after treatment is seedling establishment.
Line 38. “thrice”, do the authors mean twice?
Author Response
Authors response to Reviewer 3
Comments and Suggestions for Authors
The manuscript of Zhang and collaborators characterizes JRL gene from Populus euphratica. Their goal is to investigate the role of PeJRL in salinity tolerance, expressing the Populus gene in Arabidopsis. They propose a model where salt treatment induces the expression of this lectin that mediates regulation on ion and ROS homeostasis and represses ABA regulated genes to improve salt tolerance.
This manuscript follows a molecular and genetic approach, all the experiments show basic analysis nonetheless the data reported gives a clear message, PeJRL is involved in abiotic stress, specifically in salt stress.
The introduction is well organized and contributes to the basic knowledge to understand the results section. The experiments are well designed and executed, the amount of negative controls is appropriate and the content in the pictures is suitable, but it would be necessary to polish some of the figures. The discussion and materials/methods sections are correct even some mistakes and spelling should be addressed.
The general overview of the manuscript is positive, but cohesion is required among different parts of the manuscript as well as proofreading of the text.
Authors: First, the authors would like to thank the peer Reviewers very much for the careful review on this manuscript. Substantial revisions were made according to your valuable comments. Indicated changes were labelled with red in the text. Please see below the point-to-point explanations addressing to Reviewers’ comments and queries.
Reviewer 3: 1. Page 2 Line 40. In section 2.1, the authors comment the expression of JRL in P. euphratica leaves but they do not explain or give any idea regarding the big drop of expression showed at 24 h and why they think the plant increases the transcript production afterward.
Authors: In the revised manuscript, the expression of JRL in P. euphratica roots were also addressed according to Reviewer 2. In P. euphratica roots, PeJRL transcript level increased rapidly at 3 h after salt treatment and reached peaking level until 6 h, and then returned to pretreatment level in the following hours (Figure 1C). The drop of expression showed at leaves (24 h) and roots (12) implied the recovery of whole-plant water status after salt shock [1,2]. The following increase of PeJRL transcript in leaves at 48-72 h was presumably the result of plant response to buildup of salt ions that translocated from roots to shoots. (Lines 100-106).
Reviewer 3: 2. Page 3 Figure2. I found this figure very small and is not easy to visualize.
Authors: As suggested, the two images in Figure 2 were enlarged. We also used larger and equal font size for other images and graph labels.
Reviewer 3: 3. Page 4
a. Figure 3. The authors forgot the scale bar.
b. Line 24. The title numeration is wrong. It should be the section 2.4 instead of 2.5.
Authors: As suggested, we made indicated changes in the revised manuscript, in brief,
a. Figure 3: the scale bar was added to Figure 3 (the images showing colocalization of PeJRL and plasma membrane were removed based on comments from Reviewer 1).
b. The title numeration: the title numeration was corrected throughout the manuscript, thanks. (Lines 151).
Reviewer 3: 4. Page 5
a. Line 6. The authors forgot to explain the acronyms WT and VC. They explain them later in the manuscript but that should be done the first time they mention them which is here. They should be consistent in the nomenclature.
b. Line 9. The same problem with the section numeration, this should be section 2.5.
c. Line 15. Authors start this paragraph mentioning electrolyte leakage but without explaining the reason they are performing that analysis. I would suggest a brief introduction.
Authors: Indicated changes were made in the revised manuscript, in brief,
a. The acronyms WT and VC were explained in Figure 4 legends (Lines 162-163), and similar changes were made in all other figure legends.
b. The title numeration was corrected as 2.5 (Lines 165).
c. As suggested, a brief introduction was addressed in the text to clarify the aim to measure electrolyte leakage (Lines 173-175).
Reviewer 3: 5. Page 6
a. Figure 5A and C. Very unfortunate choice of background. The red color in figure 5C does not allow the correct visualization of Arabidopsis roots. I will encourage them to try to supply more contrasted pictures and for future experiments, I will suggest them to use a black background.
b. Figure 5A. The scale bar is very difficult to see, too thin.
c. Line 5. “survival rate”. This is not a survival rate experiment, the picture is showing seedling establishment after 10 days growing in NaCl supplemented plates. Please correct the misunderstanding appropriately in the remaining text and figures.
d. Line 8. “Each column is…” I would suggest the authors rephrasing it, for example, “Each column corresponds to the mean of…”. e. Line 12. Same problem with section numeration.
f. Line 13. I would suggest the authors explain briefly when they use reagents, such as “CoroNa Green”, in this paragraph.
g. Line 15 and 16. The authors mention figure 6A and B but there are no such letters in figure 6.
Authors: Thanks a lot for your comments. All indicated changes were made, please see below the point-to-point amendments:
a. In our previous publications, we used the red background [3,4]. Here we enlarged the images to improve the visualization of Arabidopsis roots. As you suggested, we will use a black background in future studies.
b. The bolded scale bars were inserted to in Figure 5.
c. As suggested, the inserted panel showing the survival rate was removed from the revised manuscript. Corresponding corrections were made in the text “To determine whether PeJRL affected salt tolerance during seedling establishment, seeds of WT, VC, and transgenic plants were germinated in 1/2 MS medium supplemented with increasing NaCl (75-125 mM for 10 d, Figure 5A). The two transgenic lines performed much better than WT Arabidopsis and VC at 125 mM NaCl, in terms of leaf opening and greening” (Lines 166-170).
d. Thanks a lot for your editing suggestions. Accordingly, it was rephrased as “Each column corresponds to the mean of…” in all figure legends to explain the columns.
e. The title numeration was changed to 2.6 (Line 190).
f. The aim to measure Na+ and the use of reagent, “CoroNa Green”, were addressed in this paragraph (Lines 191-194).
g. As suggested, the letter A and B were added to Figure 6.
Reviewer 3: 6. Page 7
a. Figure 6. I suggest the authors explain the left side of the figure more. “Na+” is not enough. Are they measuring levels? Which units?
b. Line 10. Same problem with section numeration.
c. Line 11. “NMT”, the acronym description is missing.
d. Line 14. “were notably higher”, a percentage would help here.
e. Lines 17-20. The authors did not connect this paragraph with the one before. Can the authors link the fluxes in the root with their decision to measure gene expression?
Authors: As you suggested, the indicated changes were made and listed as below.
a. Figure 6: The Na+ level was measured with a Na+-specific fluorescent probe. The unit represents fluorescence of the probe, a relative intensity. Therefore, it was changed to “Na+ fluorescence intensity”.
b. The title numeration was corrected (2.7) (Line 208).
c. The acronym description of “NMT” was added to the text (Lines 209-211).
d. As suggested, a percentage was added to the text (Lines 215-216).
e. According to your comment, the aim to detect gene expression was added at the beginning of this paragraph (Lines 218-219).
Reviewer 3: 7. Page 8
a. Line 15. Same problem with section numeration.
b. Line 15. “Activity and transcript...” what do the authors mean with that?
c. Line 16. Introduction to why they decide to measure the activities of anti-oxidative enzymes is missing.
Authors: Please see below the indicated changes:
a. The title numeration was corrected as 2.8 (Line 238).
b. The subtitle was changed to “Anti-oxidative enzyme activity and transcript levels of encoding genes” (Line 238).
c. As suggested, a brief introduction was addressed to explain the reason to measure the activities of anti-oxidative enzymes (Line 239-242).
Reviewer 3: 8. Page 9
a. Line 10. Same problem with section numeration.
b. Line 11. Why do the authors decide to analyze H2O2? Explain in the paragraph.
Authors: We made indicated changes as listed below:
a. The section numeration was changed to 2.9 (Line 256).
b. The explanation to analyze H2O2 was addressed in this section (Line 257-259).
Reviewer 3: 9. Page 10
a. Figure 9A. I would suggest the authors explain the title of the figure more. “H2O2” is not enough. Are they measuring levels? Which units?
b. Line 16. Same problem with section numeration.
c. Line 17. The authors do not explain why they decided to study ABA in this section.
Authors: Please see below the point-to-point response to your queries.
a. Figure 9A: H2O2 was measured with a fluorescent probe and the title was correspondingly changed to “H2O2 fluorescence intensity”.
b. The section numeration was changed to 2.10 (Line 282).
c. The explanation to analyze ABA was addressed in this section (Line 283-285).
Reviewer 3: 10. Page 11
a. Figure 10. Background problem again.
b. Figure 10. The letter size in the figure is different in each graph, I would suggest adopting one size all over the figure, if it is possible, to keep the balance.
Authors: Thank you for your careful check.
a. Figure 10: We enlarged the images to improve the visualization of Arabidopsis roots.
b. Figure 10: As you suggested, the letter size in the figure were carefully checked and amended throughout the manuscript.
Reviewer 3: 11. Page 14 Line 25. “…transgene plants”. Do the authors mean transgenic plants?
Authors: Yes. “transgene plants” was changed to “transgenic plants” in revised manuscript. (Line 323)
Reviewer 3: 12. Page 15
a. Line 1. “… enabled transgenic plants to maintain K+ under saline conditions…” What do the authors mean with K+? Is it K+ levels?
b. Line 10. “… exposure in transgene plants”. They are transgenic plants.
Authors: Indicated changes were made in the text and the revisions were list below.
a. The sentence was changed to “… enabled transgenic plants to reduce the loss of K+ under saline conditions…” (Lines 391-392)
b. Here “transgene plants” was changed to “transgenic plants” in the text (Line 400).
Reviewer 3: 13. Page 16
a. Line 25. Explain acronym “PM-specific probe”.
b. Line 30. Correct the nomenclature of the vector “PMDC85-the” to pMDC85 and the same in line 31 and 33.
c. Line 33. Correct “(Clo-0)” twice in this line.
d. Line 41 and 42. Modify sentence to, “semi-quantitative reverse-transcription PCR (RT-PCR)” and use appropriately in the remaining paragraph.
e. Line 44. Rephrase this: “DNA in isolated RNA was eliminated with RNase-free
Authors: Thank you for your careful check and valuable comments. Please see below the point-to-point explanations.
a. "PM-specific probe" means a plasma membrane-specific probe that specifically binds to the plasma membrane. However, this part was removed from the revised manuscript according to the comments from Reviewer 1 (Lines 462-470).
b. The nomenclature of the vector “PMDC85 ” was changed to “pMDC85” (Lines 472-476), thanks.
c. “(Clo-0)” was changed to “(Col-0 ecotype)” (Lines 475-476)
d. “semi-quantitative reverse-transcription PCR (RT-PCR)” was used and appropriately modified in the remaining paragraph (Lines 483-487).
e. As suggested, “The RNase-free DNase (Promega, Madison, Wisconsin, USA) was used to eliminate the DNA in isolated RNA...” was rephrased as “DNA in isolated RNA was eliminated with RNase-free DNase (Promega, Madison, Wisconsin, USA)…” (Lines 486-487).
Reviewer 3: 14. Page 17
a. Line 10. I suggest them to use MS medium supplemented with instead of “MS medium with the addition of”.
b. Line 12. “under light conditions…” do the authors mean, continuous light? Or long day, short day conditions? Please specify.
c. Line 14. What do the authors mean with “the seedlings were salinized”? The seedlings were treated with…
d. Line 42. “DCF” Is this the same reagent that in line 37? Why is now called differently? Authors: Please see below the point-to-point revisions in the text.
a. “Seeds were placed on MS medium with the addition of…” was changed to “Seeds were placed on MS medium supplemented with…”(Line 497).
b. As suggested, “under light conditions…” was specified as “under a 16 h/8 h (light/dark) photoperiod” (Line 498-499).
c. “The seedlings were salinized…” was changed to “The seedlings were treated with…”(Line 502)
d. For consistency, “DCF” was replaced with “H2DCF-DA” in the text (Line 530)
Reviewer 3: 15. Page 18
a. Line 1. Transgene plants again.
b. Line 14. “…enzyme assays”, enzymatic assays
c. Line 31. “After 3 days of incubation at 4°C”, this process is called seed stratification.
d. Line 32. “under light conditions” correct as previously suggested.
e. Line 32. Correct “survival rate” as mentioned before. This is a seed germination experiment and the measurement at 10 days after treatment is seedling establishment.
f. Line 38. “thrice”, do the authors mean twice?
Authors: Please see below the indicated changes in the text.
a. “Transgene plants” was changed to “Transgenic plants’ (Line 535)
b. As suggested, “…enzyme assays” was replaced with “ enzymatic assays” (Line 555)
c. “After 3 days of incubation at 4°C…” was changed to “After seed stratification…” (Line 572)
d. Again, “under light conditions…” was specified as “under a 16 h/8 h (light/dark) photoperiod” (Line 573).
e. Thanks a lot for the definition of seed germination and seedling establishment. Indicated changes were made in the text (Lines 569-577)
f. “thrice” was changed to “three times” (Line 584)
Authors: Finally, the authors would like to thank the reviewer again for the valuable time, careful editing, and helpful suggestions. We wish that the revision is sufficient to satisfy your positive considerations. In addition, more revisions were made according to valuable comments from Reviewer 1 and 2. For example, i) Results and Discussion were rewritten, and ii) Figures were enlarged and legends were modified.
Additional References
1. Ottow, E. A.; Brinker, M.; Teichmann, T.; Fritz, E.; Kaiser, W.; Brosche, M.; Kangasjarvi, J.; Jiang, X. N.; Polle, A. Populus euphratica displays apoplastic sodium accumulation, osmotic adjustment by decreases in calcium and soluble carbohydrates, and develops leaf succulence under salt stress. Plant Physiol. 2005, 139, 1762-1772.
2. Brinker, M.; Brosché, M.; Vinocur, B.; Abo-Ogiala, A.; Fayyaz, P.; Janz, D.; Ottow, E. A.; Cullmann, A. D.; Saborowski, J.; Kangasjärvi, J.; Altman, A.; Polle, A. Linking the salt transcriptome with physiological responses of a salt-resistant Populus species as a strategy to identify genes important for stress acclimation. Plant Physiology. 2010, 154, 1697–1709.
3. Han, Y. S.; Wang, W.; Sun, J.; Ding, M. Q.; Zhao, R.; Deng, S. R.; Wang, F. F.; Hu, Y.; Wang, Y.; Lu, Y. J.; Du, L. P.; Hu, Z. M.; Diekmann, H. K.; Shen, X.; Polle, A.; Chen, S. L. Populus euphratica XTH overexpression enhances salinity tolerance by the development of leaf succulence in transgenic tobacco plants. Journal of Experimental Botany. 2013, 64, 4225-4238.
4. Deng, S. R.; Sun, J.; Zhao, R., Ding, M. Q.; Zhang, Y. N.; Sun, Y.l.; Wang, W.; Tan, Y. Q.; Liu, D. D.; Ma, X. J.; Hou, P. C.; Wang, M. J.; Lu, C. F.; Shen, X.; Chen, S. L. Populus euphratica APYRASE2 enhances cold tolerance by modulating vesicular trafficking and extracellular ATP in Arabidopsis plants. Plant Physiol. 2015, 169, 530-548.